# Sustainable Food Security in Romania and Neighboring Countries—Trends, Challenges, and Solutions [note 1]

**DOI:** 10.3390/foods14081309

**Published:** 2025-04-09

**Authors:** Teodor Ioan Trasca, Ioana Mihaela Balan, Gina Fintineru, Jeni Veronica Tiu, Nicoleta Mateoc-Sirb, Ciprian Ioan Rujescu

**Affiliations:** 1University of Agronomic Sciences and Veterinary Medicine of Bucharest, 011464 Bucharest, Romania; gina.fintineru@usamv.ro (G.F.); jeni.tiu@usamv.ro (J.V.T.); 2University of Life Sciences “King Mihai I”, 300645 Timisoara, Romania; nicoletamateocsirb@usvt.ro (N.M.-S.); rujescu@usvt.ro (C.I.R.); 3Research Center for Sustainable Rural Development of Romania, Timisoara Branch, Romanian Academy, 010071 Bucharest, Romania

**Keywords:** sustainable food security, agricultural sustainability, food security pillars, food security indicators, regional food systems, comparative and statistical analysis, Romania and neighboring countries

## Abstract

Food security is a fundamental global challenge with significant regional implications, particularly in Eastern Europe. Romania and its neighboring countries—Bulgaria, Hungary, Serbia, Ukraine, and Moldova—face interconnected challenges related to food availability, access, stability, and utilization. This study assesses Romania’s food security in relation to its neighbors using FAO-defined indicators for each of these four pillars. The analysis is based on the latest FAO data. It applies two complementary analytical methods: comparative analysis, which evaluates Romania’s food security indicators in relation to average values in neighboring countries, and statistical significance testing, using *One-sample t-tests* and *Wilcoxon signed-rank tests* to determine whether the observed differences are statistically significant. The results indicate that Romania benefits from high food availability and a developed irrigation infrastructure but faces challenges related to severe food insecurity, economic disparities, and public health issues such as obesity. Despite a higher GDP per capita than its neighbors, food insecurity rates remain concerning, pointing to underlying socio-economic inequalities. The results highlight the need for coordinated public policies that go beyond increasing food availability to reducing regional disparities, improving equitable access to nutritious food, and promoting sustainable patterns of production and consumption. The study proposes a multidimensional and scientifically sound approach that addresses structural inequalities, strengthening the resilience of food systems and the importance of regional cooperation in Eastern Europe. These contributions add to the current international discussions on sustainable food security and provide concrete recommendations for future action.

## 1. Introduction

This research was originally presented at the 5th International Electronic Conference on Foods, where the abstract was published as part of the conference materials [1].

Food security remains one of the most pressing global challenges, with direct implications for regional stability, public health, and sustainable economic development [2,3]. The complexity of food security is particularly evident in Eastern Europe, where Romania and its neighboring countries—Bulgaria, Hungary, Serbia, Ukraine, and Moldova—face a series of interconnected challenges related to the availability, access, stability, and use of food. In these regions, food security is not only a matter of production but also of equitable distribution, adequate infrastructure, and resilience in the face of climate change, economic disparities, and political instability [4,5].

The definition of food security, based on the four fundamental pillars—availability, access, stability, and utilization—provides a robust analytical framework for assessing the situation in each country. Availability refers to the presence of sufficient food supplies, ensured through domestic production, imports, and reserves. Access encompasses economic, physical, and social factors that determine individuals’ ability to acquire food. Stability ensures continuous and reliable access to food, minimizing risks from climatic, economic, or geopolitical disruptions. Utilization relates to nutritional quality, food safety, and public health aspects that affect how food contributes to well-being. These dimensions, originally conceptualized by the FAO, are fundamental for assessing national and regional food security trends and vulnerabilities [6].

The importance of food security is closely linked to the United Nations Sustainable Development Goals (SDGs), in particular SDG 2: Zero Hunger, which aims to eradicate hunger and ensure universal access to safe, nutritious, and sufficient food. In addition, food security intersects with several other SDGs, such as SDG 1: No Poverty, which highlights the relationship between food insecurity and poverty; SDG 3: Good Health and Well-being, which promotes adequate nutrition for the health of the population; SDG 12: Responsible Consumption and Production, which emphasizes sustainable food systems; and SDG 13: Climate Action, which emphasizes the impact of climate change on agricultural production and food system stability. Given these interdependencies, food security must be addressed through a holistic and interdisciplinary approach, taking into account economic, environmental, and social factors to develop sustainable solutions [7,8].

In the last two decades, the effects of climate change on agricultural production have become increasingly evident in Romania and its neighboring countries. Specialized studies indicate a significant increase in the frequency and intensity of droughts, heat waves, and other extreme weather phenomena, with a direct impact on agricultural yields, especially in cereal and legume crops. Also, changes in the precipitation regime have affected the agricultural calendar and soil quality, accentuating the vulnerabilities of food systems, especially in rural areas with poor infrastructure. In this context, the adaptation of agriculture to new climatic conditions becomes a strategic priority for ensuring long-term food security, and the comparative analysis proposed in this study is part of this research direction [8,9].

In the specific context of the region under review, priority foods include cereals (especially wheat and maize), meat (especially pork and poultry), dairy products, and, to a lesser extent, vegetables and legumes. These products constitute the basis of the diet and reflect both food traditions and the structure of agricultural production. Agriculture plays a significant role in the economy of Romania and neighboring countries, contributing not only to GDP but also to employment, especially in rural areas. At the same time, food consumption reflects structural imbalances, such as an increased dependence on processed foods and a high share of animal products, with implications for public health and the sustainability of food systems. In this framework, an integrated approach linking production, consumption, and economic impact is essential to strengthen sustainable food security in the region [4,5].

A distinctive feature of Romania and its neighboring countries is their shared historical past as former members of the communist bloc, which profoundly shaped their economic and agricultural structures characterized by severe food insecurity driven by centralized economic policies, food rationing, and restricted access to basic commodities. The transition to a market economy has led to the diversification of agriculture and improved access to food, but regional inequalities and structural vulnerabilities persist. In Romania, the rapid liberalization of the agricultural market after 1990 led to excessive fragmentation of agricultural land as a result of its restitution to previous owners, which negatively affected productivity and investment in agricultural infrastructure. In parallel, the dismantling of the centralized food distribution system created major imbalances between urban and rural areas, especially in the underdeveloped regions of the east and south of the country, where access to markets and basic food products remained limited. These examples reflect how the economic transition, although beneficial overall, generated new forms of vulnerability in the food system.

More than 35 years after the fall of communism, these historical legacies continue to influence food security in the post-communist era [10,11]. Understanding these long-term effects is crucial for designing sustainable policies that ensure equitable access to food and economic resilience in the region.

This historical legacy was marked by the forced collectivization of agriculture, food rationing, and limited access to basic food. Agricultural policies during the communist era imposed rigid centralized planning that encouraged monocultures and neglected essential agricultural infrastructure, such as irrigation and storage [12]. After the fall of the regime, the process of restitution of agricultural land led to the extreme fragmentation of holdings and the lack of a coherent support strategy for small farmers [13,14]. These issues continue to negatively influence the efficiency of the agricultural sector, limiting the ability of countries in the region to ensure sustainable and equitable food security.

The ongoing conflict in Ukraine further exacerbates food security challenges in Eastern Europe. Ukraine, one of the world’s largest grain exporters, has faced major disruptions to supply chains, leading to significant increases in food prices and limited access to resources for import-dependent countries [12,15]. The regional impact of this conflict manifests itself in economic instability, forced population displacement, and increased vulnerabilities in food systems. Romania and its neighboring countries are directly affected by this dynamic, requiring adaptive measures to manage refugee flows and increase the resilience of the food system.

In addition to geopolitical instability, climate change plays a crucial role in shaping food security in Eastern Europe. Countries in the region, including Romania, Bulgaria, Ukraine, Serbia, Moldova, and Hungary, are increasingly facing challenges related to prolonged droughts, extreme floods, and unpredictable climate variations, which affect agricultural yields and the sustainability of food production. Romania, despite having significant agricultural potential, faces difficulties in quickly adapting to new climatic conditions, mainly due to insufficient investment in modern irrigation systems and the lack of a coherent national strategy for climate resilience in agriculture. Other countries, such as Serbia and Bulgaria, have started to develop clearer adaptation plans, but implementation remains uneven at the regional level. These disparities highlight the need for coordinated action and common policies to strengthen the resilience of food systems to future climate shocks [13,14].

The studied region presents a diverse agricultural landscape, accompanied by substantial vulnerabilities. Romania, a key agricultural producer in Europe, struggles with regional disparities in resource distribution and a lack of rural markets. Romania has significant agricultural potential, ranking among the main producers of cereals in the European Union, especially wheat, maize, sunflower, and barley. According to Eurostat and INS data, agriculture contributed approximately 4% to the GDP in 2022—the most recent year with validated and harmonized data at the European level—a percentage higher than the EU average, which underlines the strategic role of this sector. Agriculture provides jobs for around 20% of the active rural population, which is essential for socio-economic cohesion. Production systems are dual: from poorly mechanized family farms to large and integrated commercial holdings, which influences the efficiency and sustainability of the entire agri-food system [13,14].

Bulgaria and Hungary, although relatively well developed in terms of infrastructure, face challenges in terms of rising food costs and dependence on imports. Ukraine, severely affected by war, faces significant difficulties in maintaining the stability of the food system. Meanwhile, Moldova and Serbia, with less developed economies, face high rates of food insecurity and rural–urban migration [16,17].

Despite the significance of these challenges, there is a noticeable gap in the scientific literature on a unified analysis of food security in this region. Although European or global studies exist, they often fail to capture the unique historical, economic, and social characteristics of Eastern Europe. A comprehensive assessment that integrates Romania and neighboring countries in the same analytical framework is lacking, even if these countries share critical structural factors that influence food security. This absence of regional studies is a major limitation in formulating coordinated policies aimed at improving food security both at national and cross-border levels.

In addition, data fragmentation and inconsistent reporting across countries hinder an in-depth assessment of specific challenges. Many food security indicators remain underreported or inconsistently measured, complicating benchmarking of vulnerabilities and strengths across the region [18,19]. This study aims to fill this information gap by providing a comprehensive analysis based on FAO food security indicators, providing insights into both Romania’s food security status and its relative position compared to neighboring countries.

The findings of this study not only provide a new perspective on food security in Eastern Europe but also highlight the need for future research to explore in more detail the interactions between economic, political, and agricultural factors influencing food security in former planned economies.

This research aims to carry out a comparative assessment of food security in Romania and neighboring countries—Bulgaria, Hungary, Serbia, Ukraine, and Moldova—using the analytical framework based on the four pillars defined by the FAO: Availability, Accessibility, Stability, and Utilization. The main goal is to identify the strengths and vulnerabilities of each country in relation to these pillars in order to highlight the specific challenges in the region and formulate public policy recommendations. The study has a secondary objective: the proposal of strategic directions that will contribute to the development of sustainable food systems by promoting agricultural resilience, reducing inequalities, and strengthening regional cooperation. In a context marked by climate change, political instability, and economic disparities, this research contributes to a deeper understanding of the mechanisms that influence food security in the former Eastern European bloc.

## 2. Materials and Methods

### 2.1. Data Sources

To substantiate the analysis of food security in Romania and neighboring countries, a comprehensive review of relevant scientific literature was carried out using prestigious international databases, such as Web of Science, Scopus, PubMed, and ScienceDirect, alongside grey literature sources (government reports, documents of international organizations, and studies published by research institutes). In total, 58 bibliographic source works (scientific peer-reviewed papers, books, recognized institutional documents, official reports) published between 2005 and 2025 were analyzed, their selection being made based on specific keywords, such as “pillars of food security”, “food security indicators”, “sustainable food security”, “agricultural sustainability”, correlated with the regions of interest: Romania, Bulgaria, Hungary, Serbia, Ukraine, and Moldova. Additional selection criteria included conceptual relevance to the four FAO pillars of food security, applicability of the results at the regional level, and the current nature of the source (peer-reviewed papers, recognized institutional documents, or official reports). This approach allowed the identification of the most relevant publications on the factors influencing food security in the region, as well as highlighting the specific disparities and challenges in each country. To complete the analysis of scientific literature and ensure an integrated perspective on the phenomenon, documents published by international bodies were consulted, including reports by the Food and Agricultural Organization of the United Nations (FAO), World Bank, European Commission, and World Food Programme. These sources provided up-to-date statistical indicators and essential information for understanding recent trends in food security and sustainable agriculture.

This documentation stage formed the basis for the selection of FAO indicators used in the analysis and contributed to defining the research methodology, providing a solid theoretical framework for interpreting the results and formulating public policy recommendations.

These indicators, grouped into suites linked to four fundamental pillars, constitute a comprehensive data set covering multiple dimensions of food security. The analysis, therefore, focuses on the four key pillars established by the FAO: Pillar I—Availability, Pillar II—Access, Pillar III—Stability, and Pillar IV—Utilization, using the suites of statistical indicators related to each pillar. The analysis was carried out using statistical indicators specific to the four pillars of food security, according to the FAO structure. In total, 21 indicators were included in the study: four indicators for Pillar I—Availability, four indicators for Pillar II—Access, five indicators for Pillar III—Stability, and eight indicators for Pillar IV—Utilization. These indicators were selected based on the availability of data for the six countries analyzed and their relevance to the corresponding pillar. For each indicator, the most recent values reported by the FAO, updated in July 2024, were used. Since the reporting years differ depending on the indicator and the country, this variability has been mentioned as a limitation of the study in the dedicated section.

The study includes six countries: Romania and all neighboring countries, namely Bulgaria, Hungary, Serbia, Ukraine, and Moldova. The calculation methods of all indicators are established by the FAO. Data include the most recent reports available for each indicator, with reporting years varying depending on the availability of data collected by the FAO [6]. Thus, we analyzed the external data from the second and third parties, respectively from the FAO and taken over by the FAO from the reporting countries (Table 1).

The difference in reporting years will be considered in the interpretation of the results and will be discussed as a limitation of the study in Section 4.

All indicators used in this study were selected directly from the FAO Food Security Indicators Database [6]. These indicators are officially established by the FAO as part of a standardized system.

### 2.2. Methods

Two types of analyses were used to assess Romania’s food security in comparison to neighboring countries. The comparative analysis aims to identify differences between Romania and the regional average, offering a perspective on existing trends and disparities. The statistical analysis, by testing the significance of differences, determines whether these variations are statistically relevant. Combining the two approaches allows not only to describe the differences but also to validate them, contributing to a robust interpretation of the results.

#### 2.2.1. Comparative Analysis

The comparative analysis of food security in Romania compared to Bulgaria, Hungary, Serbia, Ukraine, and Moldova involved the calculation and comparison of indicator values for each pillar of food security. The aim of this analysis was to identify the main regional disparities and trends, highlighting both the strengths and vulnerabilities of each country.

The choice to compare Romania with the average of neighboring countries is methodologically based on several considerations as follows:
-*Regional relevance*. Romania and its neighboring countries have a similar historical, geographical, and economic background, having in the past agricultural systems and planned economies during the communist period. The comparison with the regional average provides a clearer perspective on how Romania is positioned in this context;-*Reducing extreme variations*. Some of these countries may have significantly different values for certain indicators (for example, Ukraine due to armed conflict or Moldova due to the size of the economy). Calculating the regional average allows for a more stable and balanced assessment;-*Generalization and public policies*. The regional average provides a useful benchmark for policymakers, highlighting not only Romania’s individual performance but also how it aligns with or deviates from overall regional trends.

The comparative analysis of food security in Romania in relation to Bulgaria, Hungary, Serbia, Ukraine, and Moldova involved the calculation and comparison of the values of the indicators related to each pillar of food security.

The calculation of the regional average of each indicator was performed using the arithmetic average formula:(1)$$\bar{X}=\frac{\sum X_{i}}{n},\;$$
where
$\bar{X}$—arithmetic average of the indicator;$\sum X_{i}$—sum of the indicator values for each neighboring country;*n*—total number of neighboring countries included in the analysis.


This analysis aims to identify the main disparities and trends in the region. The differences between Romania and the regional average of neighboring countries were examined to highlight both the vulnerabilities and strengths specific to each country. This method contributes to a more balanced interpretation of the data and facilitates the identification of common directions for improving food security at the regional level.

Not all countries analyzed reported all the indicators established by the FAO, which generated certain limitations of the available information. Additionally, in some situations, reports were expressed as values such as “<2.5” without providing exact numerical values. Therefore, where incomplete data were recorded or the “<2.5” type, they were not included in the comparative analysis to ensure the coherence of the presented conclusions.

#### 2.2.2. Statistical Analysis by Testing the Significance of Differences

To test the statistical significance of the differences between Romania’s food security indicators and the corresponding values for neighboring countries, two statistical methods were applied:-*One-sample t-test*, used for series with normal distribution;-*The non-parametric Wilcoxon signed-rank test*, used for indicators with non-normal distribution.

Since not all data may follow a Gaussian distribution, the normality of the series data was previously checked with the Shapiro–Wilk test. This fact was the basis for the decision to use the parametric variant One-sample t-test or the equivalent non-parametric Wilcoxon signed-rank test for comparisons. Although normality testing can sometimes be misinterpreted by specific tests, using Shapiro–Wilk is an action that provides more security compared to assuming the normality of the data and using parametric tests without performing an initial check.

The null hypothesis assumed that the average/median value of an indicator in Romania does not differ significantly from the regional average. The significance threshold used was α = 0.05.

Boxplots were used to illustrate the distribution and variation of food security indicators in the analyzed countries. They indicate the following:
-minimum and maximum values;-median (represented by a vertical line inside the box);

and
-average value (marked with the symbol ◊).

These visualizations make it easy to interpret the data distribution and identify any outliers or specific patterns.

Statistical analysis and visualizations were performed using *SAS OnDemand for Academics* (*SAS 9.4.*), a cloud-based statistical software widely used in research and data analysis.

In the statistical analysis for the *Pillar II—Access* indicator suite, only four indicators were included, out of an initial set of eight indicators, due to data reporting limitations and methodological considerations.

The indicator *Prevalence of undernourishment* was not analyzed because, except for Ukraine, which reported a value of 5.8%, the other countries only recorded values below an unspecified threshold, reported as “<2.5%”, making it impossible to apply statistical comparison tests. Similarly, the indicator *Number of undernourished people* was not included in the analysis because data are only available for Ukraine, while the other countries did not report this indicator.

Also, the indicators *Number of people* with severe food insecurity and *Number of people with moderate or severe food insecurity* were not included in the analysis, because they express the same reality as the indicators *Prevalence of severe food insecurity in the total population* and *Prevalence of moderate or severe food insecurity in the total population*, but in absolute terms. This expression in the form of the total number of people affected is strongly influenced by the size of the population of each country and, therefore, does not allow for correct comparisons between countries with different populations. In international comparative analysis, the use of prevalence (%) is recommended precisely to eliminate the distorting effect of population size and to allow a balanced interpretation of the food insecurity phenomenon. Thus, to ensure methodological coherence and comparability between countries, we chose to use exclusively indicators expressed in percentages. Through this approach, the study provides a detailed analysis of food security in Romania and neighboring countries, using the most recent indicators reported by the FAO. The rigorous selection of variables and the application of statistical methods allow the identification of significant differences between Romania and the average of neighboring countries, highlighting both the strengths and vulnerabilities specific to each pillar of food security. This methodology contributes to a better understanding of the factors influencing access, availability, stability, and use of food resources in the region and provides a solid basis for the formulation of policies and strategies aimed at reducing disparities and improving the sustainability of the food system.

## 3. Results

### 3.1. Results of Comparing the Value of the Pillars Indicators in Romania with the Average of Neighboring States

This section presents the descriptive differences between the indicators in Romania and the regional average without analyzing the statistical significance of these differences.

#### 3.1.1. *Availability*—Comparative Analysis of the Values of Indicators Related to Food Availability

*Food availability* is a fundamental aspect of food security, reflecting a country’s ability to provide sufficient food for its entire population. This pillar depends on factors such as domestic agricultural production, food trade balance, storage and distribution infrastructure, and agricultural market stability. The analysis of food availability in Romania and its neighboring countries is based on four key indicators: the adequacy of food energy supply, which measures whether the available resources satisfy the energy needs of the population; the share of dietary energy from cereals, roots, and tubers, which reflects the structural dependence on these food categories; average protein supply, which provides a picture of access to essential protein sources; and the average supply of animal protein, an indicator of the dietary diversity and nutritional quality (Table 2).

For *Average dietary energy supply adequacy*, Romania registers a value around 14.8% higher than the average of neighboring states. This difference highlights an increased availability of food energy in Romania, reflecting both a high domestic production capacity and the possibility of significant imports to cover the population’s demand. Although this high level is beneficial for preventing severe food insecurity, it may also indicate potential nutritional imbalances in cases of excessively energy-dense diets [20]. Compared to neighboring countries, Ukraine has the lowest value for *Average dietary energy supply adequacy*, which can be attributed to the impact of the armed conflict on agricultural production and distribution chains.

For *Share of dietary energy supply derived from cereals, roots and tubers*, Romania is 2.1% below the average of neighboring countries. This suggests a more diversified diet than in other countries in the region, where cereals and tubers constitute the main source of dietary energy. However, the high percentage for this indicator shows a significant dependence on these products, which may expose Romania to risks associated with the volatility of international cereal prices or climate change affecting agricultural production.

For *Average protein supply*, Romania exceeds the regional average by 13%, indicating a higher availability of protein resources. This result reflects robust domestic agricultural production combined with strategic imports. However, it is important to note that differences in the levels of available protein may have implications for the quality of nutrition of the population in the region.

For the *Average protein supply of animal origin*, Romania registers a value 19.4% above the average of neighboring countries. This indicator suggests a high share of animal products in the population’s diet, which can provide important nutritional benefits, such as a complete supply of essential amino acids, high bioavailability of iron and vitamin B12, thus contributing to the prevention of certain nutritional deficiencies, especially among vulnerable groups such as children or pregnant women. Ukraine and Moldova, which have the lowest values for this indicator, face economic and social challenges that limit access to these products (Figure 1).

These results highlight the fact that Romania benefits from a higher food availability than the average of neighboring countries, but it is important to address the vulnerabilities related to the relatively high dependence on cereals and the environmental impact of high consumption of animal products. Moldova, Ukraine, and even Bulgaria remain the most vulnerable countries in the region in terms of food availability, registering the low values of *Average protein supply* and *Average protein supply of animal origin,* and this context underlines the need for regionally coordinated policies to reduce disparities and improve food security.

#### 3.1.2. *Access*—Comparative Analysis of Values of Indicators Related to Food Accessibility

*Food accessibility* is a key factor in food security, influenced by economic, social, and infrastructural aspects. Even if a country has enough food, its equitable distribution and the effective ability of the population to purchase food are crucial to ensuring food security. This section analyzes indicators related to accessibility in Romania and neighboring countries, including factors such as the density of railway lines, which can influence the availability and costs of food products by facilitating their transport and distribution, gross domestic product per capita, an indicator of economic power and the ability of households to purchase food, and the prevalence of undernourishment and food insecurity, which directly reflect the challenges for population’s access to adequate nutrition.

In the analysis of indicators related to food accessibility, we did not include two indicators reported under *Pillar II—Access*, *Prevalence of undernourishment*, and *Number of people undernourished* due to limitations of available data. Reported values for the *Prevalence of undernourishment* are lower than 2.5% (<2.5) for most countries, which does not allow a detailed comparative assessment between Romania and neighboring countries. Also, for the *Number of people undernourished*, data are available only for Ukraine, which does not allow the inclusion of this indicator in a comprehensive regional analysis. The impact of missing reported values will be addressed in Section 4.3.

These limitations in data reporting can be explained by the fact that the prevalence of undernutrition in the region is relatively low, and the reporting of more detailed values is not considered necessary by international monitoring bodies. However, these shortcomings highlight the importance of more detailed and consistent data reporting to facilitate more robust cross-country analyses and comparisons. Also, the inclusion of additional indicators that better reflect food accessibility for vulnerable groups could improve the understanding of regional challenges (Table 3).

For *Railway density*, Romania registers a value 15.38% higher than the average of neighboring countries, reflecting a relatively more developed railway infrastructure that can facilitate food transport and faster access to agri-food markets. This situation gives Romania a competitive advantage in ensuring the population’s access to food, especially in rural areas. However, higher density does not necessarily mean efficiency, and the impact of infrastructure quality must be considered.

Although Romania has a moderately higher rail network density compared to neighboring countries, it is important to highlight how transport infrastructure influences food distribution and accessibility in general. A well-connected rail system facilitates the efficient transport of food products from production areas to markets, reducing transport time and costs. This can lead to lower prices, greater availability of fresh produce, and improved access to isolated or disadvantaged regions. Although this indicator does not fully reflect the quality or functionality of the rail network, its inclusion among the food security indicators established by the FAO reflects the recognition of infrastructure as a structural determinant of access to food, especially in countries with significant rural populations.

For *GDP per capita*, Romania exceeds the regional average by 56.83%, which suggests a higher economic potential for ensuring access to food by increasing the purchasing power of the population. However, this significant economic difference does not appear to be reflected in reduced food insecurity, indicating possible inequalities in income distribution or access to food. For the *Prevalence of severe food insecurity in the total population*, Romania has a value 80.20% higher than the average of neighboring countries. This result raises serious concerns about food accessibility for vulnerable categories of the population and reflects the existence of systemic challenges such as poverty, unemployment, and social exclusion.

For the *Prevalence of moderate or severe food insecurity in the total population*, Romania has a value 3.05% lower than the regional average. This result can be interpreted as a relatively better performance in ensuring minimal access to food for the majority of the population, but it does not eliminate the serious problems identified in the case of severe food insecurity.

For the *Number of people experiencing severe food insecurity*, Romania has a value 118.75% higher than the average of neighboring countries, highlighting a major disparity in the capacity to provide food for the most vulnerable categories of people, even if the economic resources available at the national level are greater than in other countries in the region.

For the *Number of severely food-insecure people*, Romania exceeds the average of its neighbors by 12.43%, highlighting that the challenges of consistent and equitable access to food remain a significant problem. This situation requires targeted measures to reduce disparities and improve social inclusion (Figure 2).

The *Prevalence of severe food insecurity in the population* and the *Prevalence of moderate or severe food insecurity in the population* are analyzed both as a percentage of the population and as an absolute number of people affected to provide a comprehensive understanding of the problem. The percentage highlights the share of the affected population in the total population, allowing direct comparison between countries with different demographic sizes. On the other hand, the absolute number reflects the effective magnitude of the problem, providing essential information for allocating the resources needed for interventions. In the case of Romania, this approach shows that, although the situation may be better in percentage compared to other countries in the region, the absolute number of affected people is significantly higher. This dual perspective contributes to a clearer identification of vulnerable groups and to the prioritization of interventions to effectively reduce food insecurity.

#### 3.1.3. *Stability*—Comparative Analysis of the Values of Indicators Related to the Stability of Food Security

*Food security stability* is an essential element in ensuring constant access to sufficient and nutritious food resources. This pillar reflects the capacity of a food system to cope with external shocks, such as climate change, political instability, or economic fluctuations, which may affect food availability. In this analysis, stability is assessed through relevant indicators, such as the percentage of arable land equipped for irrigation, the value of food imports relative to total exports, political stability, variability of food supply per capita, and the degree of dependence on cereal imports (Table 4).

For the *Cereal import dependency ratio*, Romania has a value 33.41% lower than the regional average. This suggests a higher degree of self-sufficiency in cereal production, which can be a strategic advantage in times of economic or geopolitical uncertainty.

For the *Percent of arable land equipped for irrigation*, Romania exceeds the average of neighboring countries by 379.17%. This high percentage indicates a significant capacity to manage water resources for agriculture, which contributes to the stability of agricultural production, especially in variable climatic conditions.

For the *Value of food imports in total merchandise exports*, Romania has a percentage 3.45% higher than the regional average. This suggests a relatively higher dependence on food imports, which may represent a vulnerability in the context of fluctuations in international markets.

*Political stability and absence of violence/terrorism* show that Romania is 225% more stable than the average of neighboring states. This relative stability provides a favorable context for the development of the food sector and the maintenance of food security in the long term.

For *Per capita food supply variability*, Romania registers a value 5.71% lower than the regional average. This denotes greater consistency in the supply of food resources, which contributes to the overall stability of the food system (Figure 3).

The results of the food security stability analysis indicate that Romania has a favorable position in terms of the percentage of arable land equipped for irrigation, but this advantage also highlights the need for a more efficient use of this infrastructure to increase agricultural production. At the same time, the low dependence on cereal imports and the relatively stable variability of food supply place Romania in a competitive position compared to its neighbors. However, persistent political instability in the region, particularly in Ukraine, represents a major threat to regional food security. This context highlights the importance of coordinated measures to support both Romania’s ability to maintain its food stability and the resilience of the entire region to external shocks.

#### 3.1.4. *Utilization*—Comparative Analysis of Values of Indicators Related to Food Utilization

*Utilization* focuses on how food is consumed and used efficiently by the body, as well as its quality and safety. This pillar integrates indicators related to access to water and sanitation services, the nutritional status of the population, as well as public health issues related to obesity, anemia, or low birth weight. This analysis explores how Romania positions itself compared to the average of neighboring countries in terms of these indicators, identifying strengths but also existing vulnerabilities in order to highlight priorities for effective public policies and coordinated regional interventions.

For the *Percentage of population using safely managed drinking water services*, Romania has a value 5.31% lower than the average of neighboring countries, highlighting the need for additional investments in safe drinking water infrastructure to ensure higher standards in this area. In contrast, for the *Percentage of population using at least basic drinking water services*, Romania exceeds the average of its neighbors by 3.13%, which shows that access to basic services is relatively well developed, but there are significant gaps in safe water management.

For the *Percentage of population using safely managed sanitation services*, Romania has a significantly higher value, 69.88% above the regional average, suggesting a higher level of access to safe sanitation services. However, for the *Percentage of population using at least basic sanitation* services, Romania registers a deficit of 5.38% compared to the average of its neighbors, which indicates an unequal distribution of access to such services between urban and rural areas (Table 5).

For the *Percentage of children under 5 years of age who are stunted*, Romania exceeds the regional average by 45.83%, which highlights a significant problem related to children’s nutrition and health. This aspect underlines the need for interventions to improve children’s nutritional status and reduce socio-economic inequalities. In contrast, for the *Percentage of children under 5 years of age who are overweight*, Romania registers a value 25.50% lower than the average of its neighbors, which suggests a lower risk of childhood obesity but raises questions about the overall nutritional balance.

The *Prevalence of obesity in the adult population (18 years and older)* is, for Romania, 40.03% higher than the average of its neighbors, reflecting an alarming phenomenon of obesity among adults in Romania. This trend highlights the need to promote a healthy lifestyle and policies that support food education and physical activity. Also, for the *Prevalence of anemia among women of reproductive age (15–49 years)*, Romania exceeds the regional average by 3.28%, indicating a public health problem that requires increased attention through nutritional supplements and interventions in maternal health.

For the *Prevalence of low birth weight*, Romania has a value 15.49% higher than the average of its neighbors, which highlights the need for proactive measures to monitor the health status of mothers and newborns, as well as improving access to quality prenatal care (Figure 4).

The results of the analysis indicate a mixed situation for Romania regarding *Pillar IV—Utilization*. While access to safe water and sanitation services is above the regional average, significant inequalities persist between urban and rural areas. Nutritional issues such as child undernutrition and the prevalence of obesity in adults highlight the need for strategic interventions, both nationally and regionally. These findings highlight the importance of an integrated approach that includes nutrition education, access to health services, and the reduction of socio-economic inequalities.

The indicators, *Percentage of children under 5 years affected by wasting* and *Prevalence of exclusive breastfeeding among infants 0–5 months of age*, were not included in this analysis as reported data are only available for Serbia. The lack of uniform reporting for these indicators limits the possibility of a comparative assessment at the regional level. However, their inclusion in future analyses could provide important insights into the nutritional status of children in the region.

### 3.2. Testing the Statistical Significance of the Differences Between Romania and the Average of Neighboring Countries

This section presents the statistical analysis of the differences between Romania and the regional average for each pillar of food security.

In the statistical analysis regarding the testing of the significance of the differences between Romania and the regional average, indicators presenting the same parameter in different expressions (for example, both in percentage form and in absolute numerical form) were not included. Their exclusion was determined by the fact that the results obtained would have been duplicated redundantly, providing similar information without adding additional value to the interpretation.

This methodological decision helps maintain the coherence and relevance of the analysis, avoiding unnecessary complications in data interpretation. Thus, the study focuses on indicators that provide an essential insight into the differences between Romania and the regional average without repeating already implicit conclusions. This approach allows maximizing the impact of the analysis and the clarity of the presented results.

#### 3.2.1. Availability—Statistical Analysis—Determining the Significance of the Differences Observed in Food Availability Between Romania and the Average of Neighboring Countries

For *Average dietary energy supply adequacy*, there are significant differences (*p* = 0.025) between the value of 146 in Romania and the average of 127.2 in neighboring countries. This result indicates that Romania benefits from a higher energy availability than the average of neighboring countries, which may reflect both a higher level of domestic production and a greater capacity to import and distribute food resources efficiently.

In the case of the *Share of dietary energy supply derived from cereals, roots, and tubers*, there are no significant differences (*p* = 0.839) between the value of 37 in Romania and the average of 37.8 in neighboring countries. This suggests that the structure of the Romanian diet is aligned with that of the region in terms of caloric intake from basic sources such as cereals and root vegetables.

For the *Average protein supply*, there is no significant difference (*p* = 0.125) between the value of 109.3 in Romania and the average of 90.6 in neighboring countries. This lack of significant difference suggests that Romania and its neighboring countries have comparable levels of protein supply per capita, indicating a relatively balanced availability of protein resources in the region.

For the *Average supply of protein of animal origin*, there are significant differences (*p* = 0.026) between the value of 60.1 in Romania and the average of 50.34 in neighboring countries, which indicates Romania’s greater dependence on animal protein sources, an aspect that will be discussed in detail in the next section, considering implications for sustainability, environmental impact, and economic accessibility of these products (Figure 5).

These results highlight the fact that, although Romania enjoys higher food availability than the average of neighboring countries in terms of total dietary energy and animal protein, the dietary structure remains comparable to that of the region in terms of cereal and total protein consumption.

#### 3.2.2. Access—Statistical Significance Testing—Applying Statistical Tests to Verify Whether Differences in Food Accessibility Are Statistically Relevant

For *Rail line density*, there are no significant differences (*p* = 0.875) between the value of 4.5 for Romania and the average value of 3.8 in neighboring countries. Therefore, even if in Romania, the value of this indicator is slightly higher than the average of the data series formed by the same indicator for neighboring countries, the differences are not statistically significant. This result suggests that Romania’s railway infrastructure is comparable to that of neighboring countries, without indicating a significant advantage or disadvantage in terms of accessibility of rail transport for food distribution.

For *Gross domestic product per capita*, there are significant differences (*p* = 0.044) between the value of 39873 in Romania and the average of 25423 in neighboring countries. This difference confirms a relatively more favorable economic position of Romania compared to its neighbors, which can contribute to higher access to food for the population. Given that GDP per capita is an essential indicator for the economic accessibility of food products, this result may have important implications on the ability of households in Romania to ensure their food needs compared to those in neighboring countries.

For *Prevalence of severe food insecurity in the total population (%)*, the value of 7.1 in Romania differs significantly (*p* = 0.005) from the 3.94 average value of this indicator in Romania’s neighboring countries. This difference highlights an increased vulnerability to severe food insecurity, suggesting that a larger percentage of Romania’s population faces acute difficulties in accessing food. This result requires special attention in the analyses of the socio-economic factors that contribute to this phenomenon and in the formulation of policies aimed at improving food accessibility.

For the *Prevalence of moderate or severe food insecurity in the total population*, there are no significant differences (*p* = 0.87) between the value of 19.1 in Romania and the average of 19.7 in neighboring countries. This result indicates that the level of moderate or severe food insecurity is comparable to that of neighboring countries, suggesting that the economic and social factors influencing this phenomenon are similar in the region (Figure 6).

These results show that, although Romania has a GDP per capita significantly higher than the average of neighboring countries, severe food insecurity is significantly more pronounced, which may indicate internal economic inequalities or other barriers affecting the distribution and accessibility of food in certain segments of the population.

#### 3.2.3. Stability—Statistical Analysis—Assessing the Stability of Food Security Through Statistical Tests Applied to the Differences Between Romania and the Regional Average

In the case of the *Cereal import dependency ratio*, there are no significant differences (*p* = 0.624) between the value of -81.8 corresponding to Romania and the average value of −122.8 in neighboring countries. This result suggests that Romania’s dependence on cereal imports is comparable to that of neighboring countries without indicating a statistically significant advantage or disadvantage. Although the value in Romania is higher, the statistical test does not confirm a clear difference in relation to the region.

For the *Percent of arable land equipped for irrigation*, there are significant differences (*p* = 0.004) between the value of 13.8 in Romania and the average value of 2.88 for neighboring countries. This result shows that Romania has a significantly larger irrigated area than its neighbors, which can provide a competitive advantage in the stability of agricultural production, especially in the context of climate change. This significant difference highlights the fact that Romania has invested more in irrigation infrastructure, although, compared to other European Union member states, the percentage of arable land equipped for irrigation remains low [20,21].

For the *Value of food imports in total merchandise exports (%)*, there are no significant differences (*p* = 0.908) between the value of 12 in Romania and the average value of 11.6 in neighboring countries. This result suggests that Romania and its neighbors have a similar level of dependence on food imports, without statistically significant variations. The stability of this indicator could reflect similar economic models regarding the agri-food trade balance in the region.

Nor are there significant differences (*p* = 0.120) for *Political stability and absence of violence/terrorism (index)* between the value of 0.5 in Romania and the average value of -0.4 for neighboring countries. This result suggests that Romania’s political stability does not differ significantly from that of neighboring countries, although it should be emphasized that this indicator is influenced by external factors such as regional geopolitics and ongoing conflicts. For example, Ukraine is experiencing severe instability due to the armed conflict, which influences the regional average.

Also, for *Per capita food supply variability (kcal/cap/day)*, there are no significant differences (*p* = 0.728) between the value of 33 for Romania and the average value of 35 for neighboring countries. This result indicates that the variability of food supply per capita in Romania is like that of the region, suggesting that both Romania and neighboring countries face similar challenges in terms of the stability of food supply chains (Figure 7).

The results of the statistical significance tests for this pillar show that, although Romania has a significantly larger irrigated area, the other indicators of food stability do not differ significantly from the average of neighboring countries. This finding suggests that despite targeted investments in agricultural infrastructure, Romania and its neighbors face common challenges in ensuring food security stability.

#### 3.2.4. Utilization—Statistical Analysis—Determining the Statistical Relevance of Differences Between Romania and the Regional Average

For the *Percentage of population using safely managed drinking water services*, there are no significant differences (*p* = 0.415) between the value of 0.82 in Romania and the average value of 86.6 for neighboring countries. This result suggests that access to safely managed drinking water in Romania is comparable to that of its neighbors, without indicating a significant advantage or disadvantage.

Also, for the *Percentage of population using at least basic drinking water services*, there are no significant differences (*p* = 0.095) between the value of 99 in Romania and the average value of 96 for neighboring countries. Although Romania has a slightly higher percentage of the population that has access to basic drinking water services, this advantage is not substantial enough to be considered statistically significant.

Nor are there significant differences (*p* = 0.078) for the *Percentage of population using safely managed sanitation services* between the value of 88 in Romania and the average value of 56.8 for neighboring countries. However, Romania has a much higher share of the population benefiting from safely managed sanitation services, suggesting a higher level of health infrastructure compared to most countries in the region.

Similarly, for the *Percentage of population using at least basic sanitation services*, there are no significant differences (*p* = 0.250) between the value of 88 in Romania and the average value of 93 for neighboring countries, indicating that Romania approaches the regional average in terms of access to basic health services, without registering statistically significant variations.

Compared to the *Percentage of children under 5 years of age who are stunted (modeled estimates)*, there are no significant differences (*p* = 0.608) between the value of 7.7 in Romania and the average value of 6.6 for neighboring countries, suggesting that the prevalence of stunting in children under five is relatively similar in Romania and the rest of the region, reflecting comparable nutritional and health conditions. However, compared with the *Prevalence of obesity in the adult population (18 years and older)*, there are significant differences (*p* = 0.007) between the value of 34 in Romania and the average value of 24.28 for neighboring countries, indicating a significantly higher prevalence of obesity in adults in Romania, which raises concerns about the impact of eating habits and lifestyle on public health.

Furthermore, for the *Prevalence of anemia among women of reproductive age (15–49 years)*, there are no significant differences (*p* = 0.652) between the value of 22.7 in Romania and the average value of 21.98 for neighboring countries. This suggests that the prevalence of anemia in women of reproductive age is similar in Romania and the rest of the region, without significant statistical variations.

And for the *Prevalence of low birth weight (percent) indicator*, there are no significant differences (*p* = 0.32) between the value of 8.8 in Romania and the average value of 7.62 for neighboring countries, indicating that Romania is at a comparable level to its neighbors in terms of the incidence of low birth weight, which may reflect similar socio-economic and maternal health factors across the region (Figure 8).

The results of the statistical significance tests for this pillar suggest that, although Romania has a significantly higher prevalence of obesity in adults, the other indicators do not differ significantly from the average of neighboring states. These conclusions will be analyzed in Section 4 where their implications for public health and food security at the regional level will be discussed.

## 4. Discussion

Food security is a fundamental element of economic and social stability, with direct implications for population health and sustainable development. The objective of this study was to evaluate food security in Romania compared to the average of neighboring countries—Bulgaria, Hungary, Serbia, Ukraine, and Moldova—using the four pillars defined by the FAO: availability, access, stability, and utilization. Through this approach, the analysis provides a broad perspective on Romania’s strengths and vulnerabilities in relation to neighboring countries, identifying significant disparities and issues that require strategic intervention.

This assessment, essential in the context of current challenges generated by climate change, economic inequalities, and geopolitical instability, directly influences food security at global and regional levels. Comparing Romania with the average of neighboring countries not only highlights the specific characteristics of each country but also identifies common trends and possible directions for coordinated intervention.

By integrating rigorous statistical methods, the research contributes to a deeper understanding of the determinants of food security, providing a solid basis for formulating sustainable food policies aimed at ensuring equitable access to food and efficient use of available resources, given that the reduction of regional disparities is a strategic objective both at national and European level.

### 4.1. Comparative Analysis—Main Findings

#### 4.1.1. Pillar I—Availability

The results of the comparative analysis indicate that Romania benefits from a high level of food supply compared to the average of neighboring countries. This situation reflects a solid agricultural capacity and diversified production, contributing to ensure a stable food supply. However, such increased availability has significant implications for both sustainability and population health.

Although the *Average dietary energy supply adequacy* in Romania is significantly higher than the average of neighboring countries, this does not necessarily reflect an optimal sustainable food system. A high energy supply, especially when associated with high consumption of animal products and ultra-processed foods, can lead to overconsumption, nutritional imbalance, or increased levels of food waste. These patterns can have negative environmental implications, including higher greenhouse gas emissions and inefficient land use. Therefore, energy availability should be interpreted in relation to food quality, dietary diversity, and sustainability considerations.

One of the relevant aspects is diet diversity. Romania records a higher value for indicators *Average dietary energy supply adequacy* and *Average protein supply*, indicating a higher consumption of food and protein compared to the average of neighboring countries. In particular, the high level for the indicator *Average supply of protein of animal origin* suggests a significant intake of animal products in the diet. This can be nutritionally beneficial, but it also raises concerns about the ecological impact of food production. Livestock farming involves intense consumption of natural resources, high emissions of greenhouse gases, and soil degradation. Thus, although Romania has an advantage in terms of food availability, careful management of the sustainability of the food system is necessary [22,23].

This protein intake is supported by a dual agricultural system in Romania, which combines small-scale family farms, mainly oriented towards self-consumption, with large-scale commercial farms, vertically integrated and focused on supplying the domestic market and exports [9,10].

Another important aspect is addiction to certain food categories. Although Romania has a superior food energy supply, a significant part of it, included in the indicator *Share of dietary energy supply derived from cereals, roots, and tubers*, makes it vulnerable to production fluctuations caused by climate change, drought, or other extreme phenomena. This dependence indicates a pressing need to diversify supply sources and adopt more resilient agricultural practices that reduce the risks associated with climatic and economic variations.

In conclusion, the increased availability of food in Romania represents a clear advantage compared to the average of neighboring states, but this situation also imposes a series of challenges related to sustainability, resource management, and maintaining the nutritional balance of the population. Agricultural and food policies must address these issues to ensure not only sufficient food production but also a sustainable and resilient food system.

#### 4.1.2. Pillar II: Access

The comparative analysis of food accessibility in Romania and the average of neighboring countries reveals a strong contrast between economic performance and the level of food insecurity. Romania is significantly above the average of neighboring countries in terms of *Gross domestic product per capita, PPP*, which could suggest a greater economic capacity of the population to access food. The data show, however, that this economic advantage does not translate into a proportional reduction in food insecurity, with Romania registering a significantly higher *Prevalence of severe food insecurity in the total population* than the average of neighboring countries. This discrepancy indicates that GDP per capita is not always a sufficient indicator to assess real access to food.

A possible explanation for this situation is the unequal distribution of income and social disparities in Romania. Although average per capita income is higher, there are significant segments of the population that face high food costs, low wages, and limited access to the resources needed for adequate nutrition. In this context, vulnerable groups—such as people in rural areas, low-wage workers, and certain disadvantaged categories—are more exposed to the risk of food insecurity.

Another relevant aspect is the difference between severe and moderate food insecurity. While the indicator *Prevalence of moderate or severe food insecurity in the total population* does not show significant differences between Romania and the average of neighboring countries, the level of *Prevalence of severe food insecurity in the total population* is alarmingly high. This suggests that although a large segment of the population has relatively stable access to food, there is a sizable group that suffers from extreme food insecurity, with major impacts on their health and well-being.

Also, a determining factor in food accessibility is infrastructure and connectivity. Romania records a slightly higher value for the indicator *Rail line density*, but the difference with the average of neighboring countries is not significant. This suggests that although there is a transport network that facilitates food distribution, it is not sufficiently well developed to reduce regional inequalities in access to agri-food markets and fresh food.

This paradox can be explained by the deep income inequalities that persist between regions and social groups in Romania. According to national and European statistics, rural areas in Romania are still disproportionately affected by poverty, and many low-income households struggle with limited access to nutritious and affordable food [16,24]. In this context, a high GDP per capita can mask internal disparities where economic growth does not benefit all categories of the population equally. In addition, high inflation rates and rising food prices in recent years have disproportionately affected vulnerable groups, amplifying the risks of food insecurity despite macroeconomic progress [23,25]. As such, economic indicators need to be interpreted considering social equity and access to resources at the household level.

#### 4.1.3. Pillar III—Stability

Romania presents a relatively high level of food security stability compared to the average of neighboring countries, but this stability is influenced by agricultural, economic, and geopolitical factors that can amplify or diminish long-term risks.

One of the most notable advantages of Romania is arable land equipped for irrigation, quantified by the indicator *Percent of arable land equipped for irrigation*. Romania significantly exceeds the average of neighboring countries, which indicates a better-developed agricultural capacity and a greater potential for resilience to climate variability. This characteristic is essential for maintaining stable agricultural production, especially in the context of climate change, which is having an increasing impact on agriculture in Eastern Europe [21,26].

Despite this agricultural advantage, Romania faces economic vulnerabilities related to dependence on food imports [27]. Although the differences for indicator *Value of food imports in total merchandise exports* are not statistically significant, the fact that this indicator is slightly higher than the average of neighboring countries suggests a moderate dependence on imports for certain product categories. This can become a problem in situations of economic or geopolitical crisis, when price fluctuations and supply chain disruptions could affect the availability of essential food products [28].

Another critical aspect is political stability, measured by *Political stability and absence of violence/terrorism*. Romania has a higher score than the average of its neighboring countries, but this difference is not statistically significant. Although Romania benefits from a more stable political environment compared to countries such as Ukraine or Moldova, risks related to economic instability, social tensions, and regional geopolitical influences can affect food security in the long term [29,30].

Regarding the variability of food supply, *Per capita food supply variability* shows that Romania has a relatively similar fluctuation to the average of neighboring countries. This indicates that, although there is a relatively stable agricultural production, Romania is not completely insulated from the volatility of international agri-food markets and possible economic crises.

Conversely, dependence on cereal imports, measured by the *Cereal import dependency ratio*, is significantly lower in Romania than the average of neighboring countries. This aspect indicates an important advantage in ensuring long-term food security, reducing vulnerability to fluctuations in international cereal prices and possible trade restrictions imposed in crisis situations.

#### 4.1.4. Pillar IV—Utilization

The comparative analysis indicated significant differences in the use of food resources and their impact on public health in Romania, compared to the average of neighboring countries. One of the most alarming aspects identified is the *Prevalence of obesity in the adult population (18 years and older)*, where Romania registers a significantly higher value than the average of its neighbors. This trend can be explained by dietary changes in the last decades, characterized by an increased consumption of processed foods and products with high caloric density, corroborated with an increasingly sedentary lifestyle [4,11]. This situation draws attention to the need for more effective prevention strategies and nutritional policies that promote a balanced diet and an active lifestyle.

The high prevalence of obesity among the adult population in Romania reflects not only a nutritional imbalance but also an inefficient use of food resources, with significant long-term implications for both public health and the sustainability of the food system. From a food security perspective, excessive calorie consumption leading to obesity can be interpreted as metabolic food waste—a form of food waste that occurs when food is consumed more than nutritional requirements, without contributing to the health of the body or to the sustainable use of resources. In the long term, obesity favors the increase in the incidence of chronic non-communicable diseases, such as diabetes, cardiovascular diseases, or certain types of cancer, which generates additional pressures on the health system and reduces economic productivity. In addition, excessive consumption of ultra-processed foods, often associated with low nutritional quality, affects not only individual health but also the ecological footprint of the food system. Therefore, reducing obesity is not only a public health objective but also an essential component of strengthening food security and sustainability, which requires integrated policies focused on balanced nutrition, food education, and equitable access to health products for all social categories [27].

For the *Percentage of population using safely managed drinking water services*, Romania has a lower value than the average of neighboring countries, which may raise concerns about the safety of drinking water and its impact on public health. Access to quality water is essential not only for hydration but also for food hygiene, preventing the transmission of waterborne diseases and reducing the risks associated with malnutrition. However, Romania is above the average of neighboring countries for the *Percentage of population using at least basic drinking water services*, which indicates relatively good access to minimally safe water sources, but with the need to improve their quality management.

At the same time, the *Percentage of children under 5 years of age who are stunted* is higher in Romania than the average of neighboring countries, suggesting possible nutritional deficiencies in the first years of life, which can have negative effects on cognitive and physical development in the long term. This may indicate socio-economic disparities or problems of access to a balanced diet in certain vulnerable categories of the population. In contrast, the *Percentage of children under 5 years of age who are overweight* is lower in Romania than the average of neighboring countries, which could signal a different distribution of nutritional risks between the analyzed countries.

For the *Prevalence of anemia among women of reproductive age (15–49 years)*, Romania records values like the average of neighboring countries, which suggests that this aspect remains a regional challenge and requires common intervention strategies to reduce iron deficiencies and improve the health of mother and child.

Thus, Romania presents a series of challenges and advantages in the field of food use and safety, and public policies must aim both at combating obesity and malnutrition, as well as improving water quality and access to health services to ensure sustainable and nutritionally balanced food security.

A particularly important aspect identified in this analysis is the high prevalence of obesity among the adult population in Romania, which significantly exceeds the average of neighboring countries. This phenomenon reflects not only a nutritional imbalance but also an inadequate use of food resources, with major implications for public health. In the long term, obesity contributes to the increase in the incidence of chronic non-communicable diseases, such as diabetes, cardiovascular diseases, and certain types of cancer, which leads to additional pressure on the health system and reduced economic productivity. In addition, ultra-processed foods, frequently consumed in excess, affect not only health but also the sustainability of the food system through their negative impact on the environment. Therefore, reducing obesity is not only a public health objective but an essential component of strengthening food security, which requires integrated policies that promote balanced nutrition, nutritional education, and access to healthy foods for all social categories.

Another important and often underestimated, but essential aspect in assessing food security, is food loss and food waste, phenomena that cross-cuttingly affect all four pillars of food security. In the context of Eastern European countries, food losses occur mainly in the initial stages of the food chain—from production and post-harvest to processing and distribution—being generated by inadequate infrastructure, lack of storage and preservation capacities, or restrictive commercial standards. In parallel, food waste manifests itself in the final phase, at the consumer level, being frequent in households, in the HoReCa sector, and in public institutions, against the background of inefficient eating behaviors, lack of nutritional education, and inadequate resource management [31]. These dysfunctions not only limit the effective availability of food but also lead to the waste of natural resources, aggravate the climate impact of the food system, and amplify the paradox of food insecurity in regions where resources are apparently sufficient [32,33,34,35]. In this sense, the prevention and reduction of food losses and waste must become a priority direction for sustainable food security policies in a coherent and integrated approach at the regional level.

### 4.2. Results of Statistical Significance Testing

Testing the significance of the differences between Romania and the average of neighboring countries provided a clear perspective on the main aspects that differentiate food security in Romania from the region. Within each analyzed pillar, certain indicators showed significant differences, emphasizing both Romania’s specific advantages and vulnerabilities in relation to its neighbors.

One of the most relevant results was observed for *Average dietary energy supply adequacy*, where Romania recorded a value significantly higher compared to the average of neighboring countries (*p* = 0.025). This fact confirms a high availability of food in Romania, which suggests that the country has a higher food production and supply capacity towards its neighbors. However, this oversupply must also be analyzed from the perspective of its impact on the sustainability and efficiency of the food system because excessive or inefficient consumption of resources can generate risks of food loss and waste.

For *Gross domestic product per capita*, statistical tests indicated a significant difference (*p* = 0.044), confirming that Romania has a higher GDP per capita compared to the average of neighboring countries. However, this economic advantage is not reflected in improved access to food for all social categories, given that the indicator is significantly higher in Romania (*p* = 0.005) than in the neighboring countries. This discrepancy suggests that social and economic inequalities play an important role in food accessibility, emphasizing the need for policies to support vulnerable groups and measures to reduce social exclusion in food supply.

Another important result of the analysis was observed in the percentage of arable land equipped for irrigation, where Romania presented a significant positive difference (*p* = 0.004) compared to the average of neighboring countries.

This indicates an important agricultural advantage, with Romania having a greater capacity to irrigate arable land, contributing to a more productive agriculture and less vulnerability to extreme weather conditions. However, this advantage must be maintained and consolidated through continued investments in irrigation infrastructure and modern agricultural technologies to increase the resilience of the agricultural system to climate change.

For the *Prevalence of obesity in the adult population (18 years and older)*, Romania has a significantly higher value (*p* = 0.007) compared to the average of neighboring countries. This result indicates an increased risk of metabolic disorders and diseases associated with an unhealthy lifestyle, emphasizing the importance of nutrition education programs and the promotion of a healthy lifestyle. Without effective interventions, this trend could lead to an increase in the costs of the public health system and a decrease in the quality of life of the adult population.

The results of the statistical significance test indicate deep-rooted structural factors influencing Romania’s food security. On the one hand, there are clear advantages, such as high food availability and superior agricultural infrastructure, but on the other hand, economic inequalities and unequal access to resources continue to affect a significant part of the population. Thus, future policies must focus on balancing these differences through measures to support vulnerable groups, improve access to healthy food, and increase the sustainability of the agri-food system.

### 4.3. Study Limitations and Shortcomings

A limitation of this study is related to the availability and consistency of data. Certain indicators were not included in the analysis due to the lack of systematic reporting in all countries analyzed. For example, the indicator *Prevalence of undernourishment (%)* was only reported with a generic value of “<2.5%” for most countries, which made it impossible to include it in the comparative analysis or test for statistical significance. Similarly, the *People undernourished indicator* was not analyzed because data for this parameter were only available for Ukraine, and the lack of values for the other countries would have led to an incomplete and inconclusive analysis. These limitations highlight the importance of improving data collection and reporting methods internationally so that future studies can include a wider range of indicators and provide a more detailed picture of food security in the region.

An important methodological aspect of this study is the difference between the reporting years of the *Food Security Indicators*, determined by the variable availability of data provided by the FAO. This temporal discrepancy constitutes a limitation of the analysis, as the values reported for certain countries correspond to different periods, which may influence direct comparisons between Romania and the average of neighboring countries.

This problem is frequently encountered in food security studies, where the frequency of data collection and updating of indicators varies depending on national reporting policies and the capacity of institutions to provide statistical information [27,28,29,30]. For example, some key indicators, such as the *Prevalence of food insecurity* or *Cereal import dependency ratio*, are collected and published at different time intervals, which may affect the accuracy of regional trends compared in this study.

To minimize the impact of this limitation, the analysis was constructed to use the most recent data available for each indicator, ensuring the most accurate representation of the current reality. Therefore, the interpretation of the results must be performed in the context of this variability of reporting years, considering that some observed differences may reflect both distinct economic and social realities and different moments of data collection.

This issue is clarified by the direct response from the FAOSTAT team following a formal request submitted by the authors. FAO specialists confirmed that the data published on the FAOSTAT platform were last updated in July 2024, and the latest available values refer to the year 2023. Furthermore, for many indicators, especially those provided by other organizations such as UNICEF and WHO, the FAO publishes the latest and most reliable version available, which may refer to previous years and multi-year averages. As such, the structure and format of these indicators (including temporal variations) are not the result of the authors’ choice but reflect the highest level of data accessibility possible at present. This clarifies why certain indicators refer to intervals (e.g., 2020–2022 or 2021–2023), while others are values for a single year, and supports the decision to use them as they are, for consistency with international standards.

In this context, future studies could benefit from more homogeneous data and more frequent collection of indicators, which would contribute to greater precision in the assessment of food security at the regional level.

Thus, this research highlights the need to harmonize the data reporting process within the analyzed countries so that food security policies are based on comparable, updated, and relevant information for strategic decisions in the agri-food sector [36,37,38].

From a methodological point of view, the current study focused on comparing the average values of neighboring countries with those of Romania without considering the internal distribution of these indicators within each country. In future studies, a more detailed analysis at the subnational level, both in Romania and neighboring countries, could highlight significant regional disparities and allow the formulation of more targeted public policies. Also, the use of econometric models investigating the causal relationships between various socio-economic factors and food security could bring additional insights into the mechanisms that influence these disparities [39].

Despite these limitations, the study offers a clear and well-founded perspective on food security in Romania and the region, contributing to a better understanding of the specific strengths and vulnerabilities of this geopolitical and economic context.

### 4.4. Policy Implications and Recommendations

The results of this study highlight the need for coordinated public policies at the regional level aimed at reducing the disparities between Romania and neighboring countries in terms of food security. Based on the four analyzed pillars– availability, accessibility, stability, and utilization—some specific recommendations can be formulated, aimed at contributing to improving food security in a sustainable and equitable way (Figure 9).

Regarding *food availability*, the data indicate that Romania benefits from a high level of food energy and protein supply, but this advantage should be supported by measures to diversify agricultural production and reduce the environmental impact of the high consumption of animal products. Public policies should encourage sustainable agricultural technologies, agroecological practices, and the development of food processing infrastructure, thus reducing food waste and ensuring a more equitable distribution of resources [18,41].

In terms of *food accessibility*, the study highlighted a major discrepancy between Romania’s high level of GDP per capita and alarming rates of severe food insecurity. This situation suggests that economic factors are not the only determinants of food access and that social inequalities and income distribution must also be considered. In this regard, governments should implement support programs for vulnerable groups, subsidies for basic foods, and policies that improve access to markets and rural infrastructure.

For *food security stability*, Romania has advantages in terms of agricultural infrastructure, but geopolitical and economic vulnerabilities in the region can negatively influence food security in the long term. It is recommended to increase the resilience of agriculture by investing in irrigation, diversifying import and export sources, and strengthening regional partnerships to avoid excessive dependence on certain markets [19,41]. Food security and crisis management measures should also be strengthened so that states can respond effectively to economic or climate shocks.

Regarding *food use*, the high level of obesity and other public health problems in Romania, compared to the average of neighboring countries, highlights the need for policies on nutrition education and promotion of a healthy lifestyle. Awareness campaigns on balanced nutrition, regulation of the nutritional content of processed products, and improved access to public health services can help combat these problems [42,43,44,45]. Also, the expansion of drinking water networks and sanitation infrastructure in rural areas could reduce the risks of malnutrition and disease, thus contributing to the overall improvement in the use of food resources.

To reduce disparities and improve food security in the region, an integrated approach combining economic, social, and environmental interventions is essential [46,47,48]. Cooperation between countries in the region, supporting food security research, and developing national and cross-border adaptation strategies to climate change and economic crises could strengthen the resilience of food systems in Eastern Europe [49].

Although the importance of regional cooperation in improving food security is evident, a pertinent analysis detailing the current state of collaboration between Romania and its neighboring countries has not yet been developed. This dimension is complex, influenced by political, institutional, and economic factors, and should be explored in future research to better understand the mechanisms, opportunities, and barriers that influence regional integration in food security policies. Therefore, public policies must focus not only on increasing food production but also on ensuring equitable access, maintaining stability, and promoting sustainable use of food resources. This multidimensional approach is essential to combat severe food insecurity and ensure a sustainable and resilient food system across the region.

The implementation of these recommendations, however, depends on several contextual factors, such as the institutional capacity of each country, the available financial resources, political priorities, and the degree of collaboration between public and private actors. Also, barriers such as poor infrastructure, resistance to change, or territorial inequalities may hinder the uniform application of the proposed policies, requiring flexible adaptations to the realities of each country analyzed.

Moreover, although this analysis focuses on Romania and its neighboring countries, many of the challenges identified—such as economic inequalities, insufficient rural infrastructure, and vulnerability to climate change—are also found in other regions of the world, including Latin America, Asia, and North America. In Brazil, for example, the “Fome Zero” (Zero Hunger) program integrated food assistance policies, school feeding programs, and support for small farmers, contributing significantly to hunger reduction and rural development. In India, public food distribution systems (PDSs) have been widely implemented to ensure access to food for the vulnerable population, despite systemic difficulties. In the United States, initiatives such as SNAP (Supplemental Nutrition Assistance Program) and community-supported agriculture have been developed to combat food deserts and nutritional inequalities [50]. These examples demonstrate that the complex food security challenges require multifaceted policy responses, adapted to the specificities of each context. Therefore, strengthening international cooperation and promoting knowledge exchange can support the development of adaptive public policies that increase the resilience of food systems, not only in Eastern Europe but also globally. In this broader perspective, alignment with SDG 17: Partnerships for the Goals becomes essential, as sustainable food security can only be achieved through strategic collaborations between countries, institutions, and communities.

## 5. Conclusions

This study provides a comprehensive and integrated assessment of food security in Romania compared to its neighboring countries, Bulgaria, Hungary, Serbia, Ukraine, and Moldova, using the four-pillar framework developed by the FAO: Availability, Access, Stability, and Utilization. The results provide an in-depth understanding of the strengths, weaknesses, and disparities characterizing food systems in this historically and geopolitically complex region.

From the perspective of *Pillar I—**Availability*, Romania presents higher values for indicators such as *Average dietary energy supply adequacy* and *Average protein supply*, reflecting a strong domestic agricultural capacity and sufficient food production. However, this advantage is offset by a strong reliance on animal-based proteins, indicated by a high *Average supply of protein of animal origin* and a significant share of calories derived from cereals, roots, and tubers. These consumption patterns suggest a limited dietary diversity and raise concerns about environmental sustainability due to the ecological footprint of livestock farming. Romania’s production system needs to evolve to prioritize more sustainable practices, crop diversification, and the promotion of plant-based dietary models.

In terms of *Pillar II—**Access*, the paradox between a high GDP per capita and a very high *Prevalence of severe food insecurity* highlights the persistence of deep economic inequalities and the exclusion of vulnerable social groups. Despite a relatively well-developed rail infrastructure, the density of railway lines is above the regional average, and access to food remains unequal across different territories. Structural factors such as low income in rural areas, inadequate social safety nets, and unequal distribution of resources prevent economic progress from translating into equitable access to food. These findings support the urgent need for redistributive policies, investments in rural development, and inclusive food assistance programs targeting marginalized populations [51,52,53].

Regarding *Pillar III—Stability*, Romania stands out with a higher share of arable land equipped for irrigation, which enhances agricultural resilience and supports food production in the context of climate variability. At the same time, however, the country faces challenges such as its moderate dependency on cereal imports, fluctuating political stability scores, and exposure to regional geopolitical shocks, especially those stemming from the ongoing conflict in Ukraine. Although this study did not directly assess climate change, it incorporated proxy indicators such as food supply variability and irrigation capacity, providing a partial lens into systemic stability. In future research, a more explicit incorporation of climate risk models would strengthen the ability to anticipate vulnerabilities and guide adaptive policies.

Regarding *Pillar IV—Utilization*, the study reveals worrying trends in Romania’s public health status, in particular the high *Prevalence of obesity in the adult population*. This condition indicates not only dietary imbalance but also a form of metabolic food waste—an inefficiency resulting from excessive intake in relation to nutritional needs. Overconsumption contributes to the increasing burden of non-communicable diseases, such as diabetes and cardiovascular disorders, increasing pressure on health systems and negatively impacting economic productivity. Together with disparities in access to sanitation and drinking water in rural areas, this situation highlights the need for a multi-sectoral approach that promotes nutritional education, equitable access to healthy foods, and improvement of basic infrastructure.

A notable limitation encountered during the research process was the lack of comprehensive data reporting. The indicator *Prevalence of undernourishment* was reported generically as “<2.5%” for most countries, rendering it unusable for meaningful comparisons or statistical tests. Similarly, the indicator *Number of undernourished people* was only available for Ukraine. These inconsistencies limit the granularity of cross-country assessments and emphasize the need for harmonized data collection and timely updates by international and national institutions.

In parallel, the importance of regional cooperation has emerged as a critical component of ensuring long-term food security. While this study highlighted its significance, it did conduct an in-depth assessment of the current state of collaboration between Romania and its neighboring countries. The lack of accessible data and limited public reporting on cross-border food security policies remain major barriers [54,55,56]. This gap highlights the need for future research that explores the institutional mechanisms, strategic frameworks, and operational challenges of regional coordination in Eastern Europe [7,57].

Another key element discussed in the study is food waste, a dimension often neglected in traditional food security assessments. Although only briefly touched upon, the paper introduces both classical food waste (at the consumption level) and metabolic food waste as critical issues in Romania and the entire region. In the HORECA sector, as well as in households, significant amounts of edible food are thrown away due to poor planning, inefficient purchasing behavior, or cultural norms. Tackling food waste along the entire supply chain, from production to consumption, can help improve resource efficiency, reduce environmental stress, and enhance the sustainability of the food system [7,57].

Based on the findings of this study, a number of policy recommendations can be formulated as follows:Promote nutrition-sensitive educational programs and public health campaigns to address obesity and prevent malnutrition;Expand investments in transport and irrigation infrastructure, especially in disadvantaged rural areas, to improve physical access to food and agricultural productivity;Strengthen social safety nets and implement targeted food assistance programs for vulnerable groups;Encourage regional cooperation on food security monitoring, early warning systems, and knowledge exchange;Develop national strategies to reduce both physical and metabolic food waste, integrating them into broader sustainability plans;Harmonize data collection efforts across the region to ensure comparable and actionable food security indicators.

The results of this study also strongly align with the Sustainable Development Goals (SDGs). While SDG 2: Zero Hunger remains the central reference point, the findings also relate directly to SDG 1: No Poverty, as economic inequalities hinder food access; SDG 3: Good Health and Well-being, due to public health challenges associated with diet and obesity; SDG 6: Clean Water and Sanitation, highlighting rural infrastructure deficits; SDG 12: Responsible Consumption and Production, through the focus on food waste and overconsumption; and SDG 13: Climate Action, due to vulnerability to climate-induced agricultural shocks.

In conclusion, Romania benefits from robust agricultural potential and favorable macroeconomic indicators, but its food security remains fragile due to persistent access disparities, environmental pressures, and insufficient infrastructure. This research responds to a visible gap in the literature by offering the first comprehensive, comparative analysis of food security in Romania and its neighboring countries based on the FAO’s four-pillar model. It provides actionable insights and lays the foundation for future studies that could incorporate subnational data, climate risk scenarios, and governance dynamics to build more resilient food systems in Eastern Europe. Thus, food security cannot be approached as an isolated challenge but must be understood as an integrative element in broader sustainable development strategies [58]. Romania and its neighboring countries need coordinated policies that address not only the immediate aspects of food access and availability but also the structural factors that influence long-term stability and efficient use of resources. Only through a multidimensional approach, which includes economic development, environmental protection, and social equity, can a sustainable future for regional food security be ensured.

## Figures and Tables

**Figure 1 foods-14-01309-f001:**
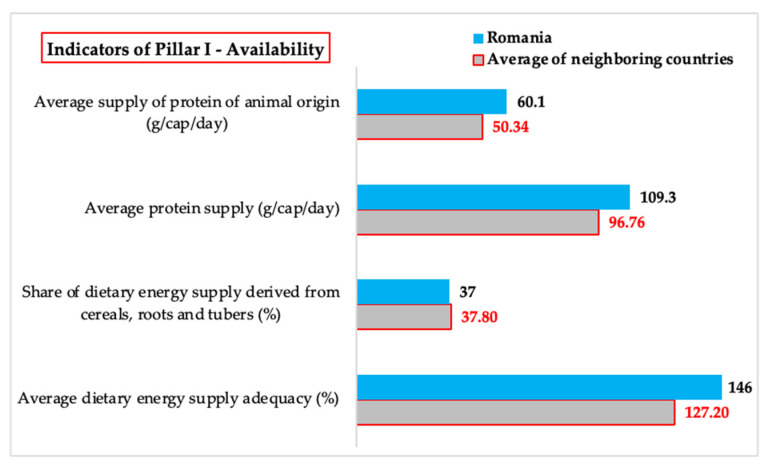
Comparison of *Pillar I—Availability* indicators.

**Figure 2 foods-14-01309-f002:**
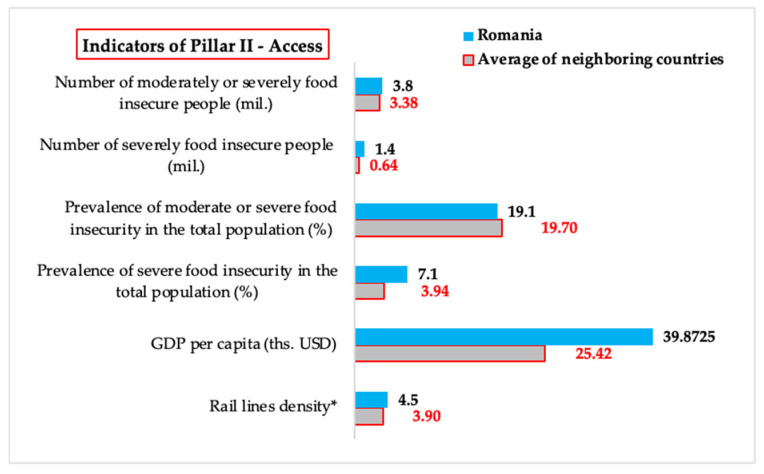
Comparison of *Pillar II—Access* indicators. * For Rail line density, there are no significant differences (*p* = 0.875) between the value of 4.5 for Romania and the average value of 3.9 in neighboring countries.

**Figure 3 foods-14-01309-f003:**
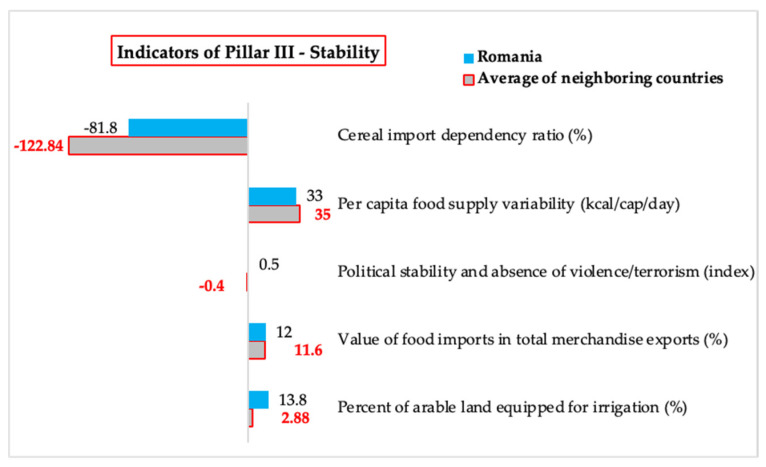
Comparison of *Pillar III—Stability* indicators.

**Figure 4 foods-14-01309-f004:**
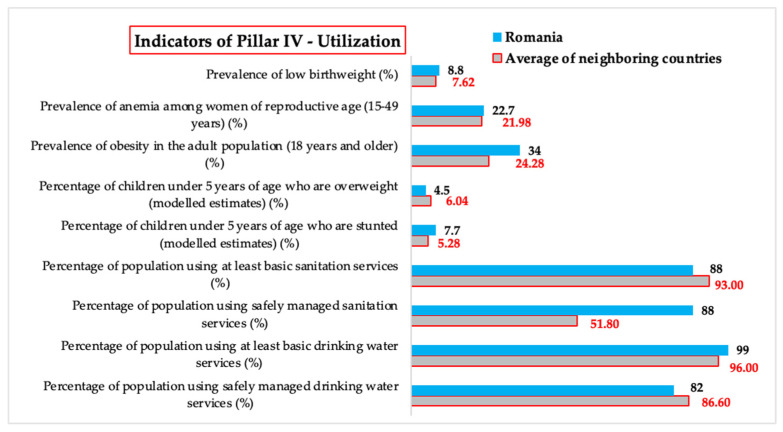
Comparison of *Pillar IV—Utilization* indicators.

**Figure 5 foods-14-01309-f005:**
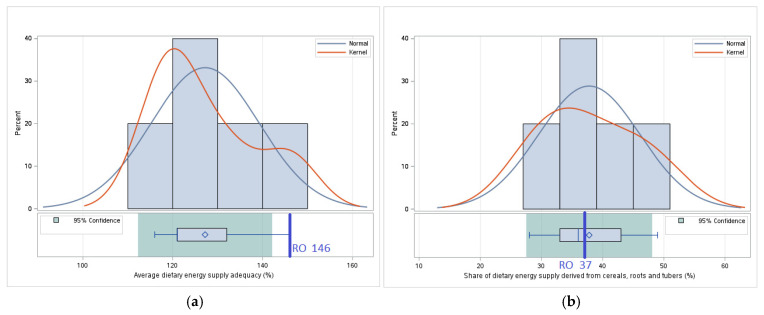
Distribution of values for *Pillar I—Availability* indicators suite in Romania’s neighboring countries, respectively, the value of the indicator in Romania (RO): (**a**) Average dietary energy supply adequacy (%); (**b**) share of dietary energy supply derived from cereals, roots, and tubers (%); (**c**) average protein supply (g/head/day); (**d**) average supply of protein of animal origin (g/cap/day).

**Figure 6 foods-14-01309-f006:**
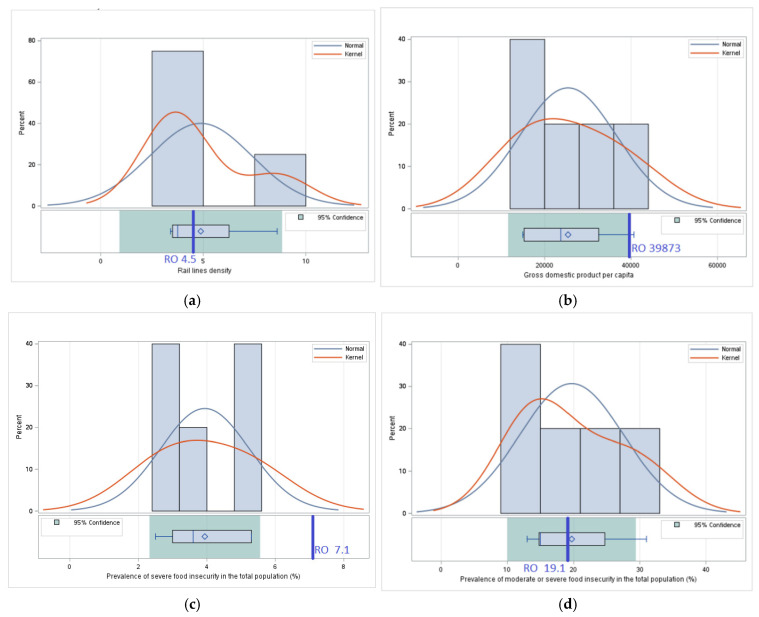
Distribution of values for *Pillar II—Access* indicator suite in Romania’s neighboring countries, respectively, the value of the indicator in Romania (RO): (**a**) Average dietary energy supply adequacy (%); (**b**) share of dietary energy supply derived from cereals, roots, and tubers (%); (**c**) average protein supply (g/head/day); (**d**) average supply of protein of animal origin (g/cap/day).

**Figure 7 foods-14-01309-f007:**
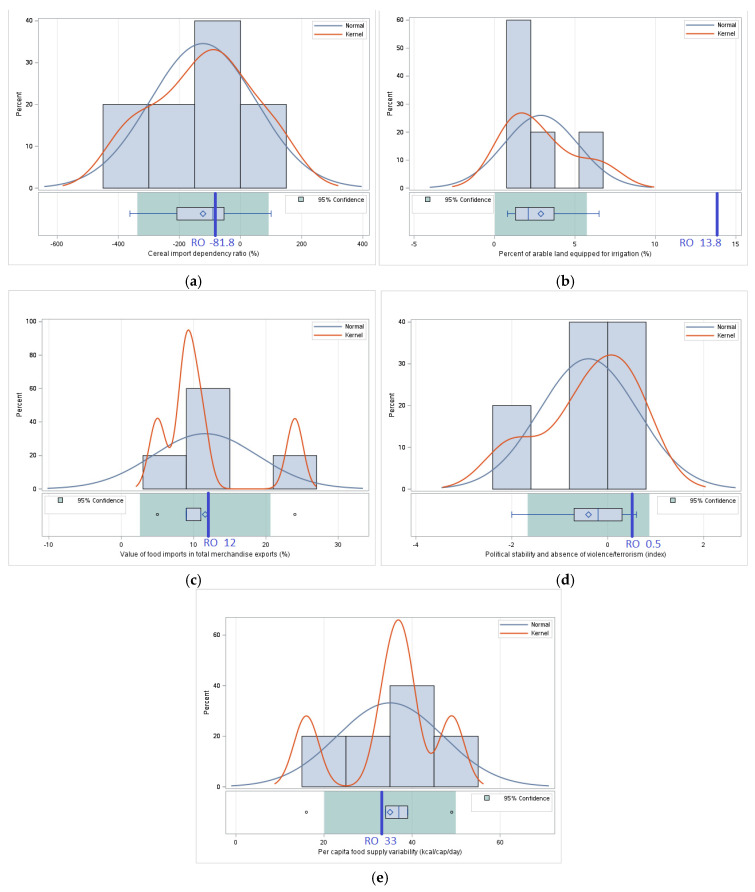
Distribution of values for *Pillar III—Stability* indicator suite in Romania’s neighboring countries, respectively, the value of the indicator in Romania (RO): (**a**) Rail lines density (total route in km per 100 square km of land area); (**b**) Gross domestic product per capita, PPP, (constant 2017 international $); (**c**) prevalence of severe food insecurity in the total population (%); (**d**) prevalence of moderate or severe food insecurity in the total population (%); (**e**) per capita food supply variability (kcal/cap/day).

**Figure 8 foods-14-01309-f008:**
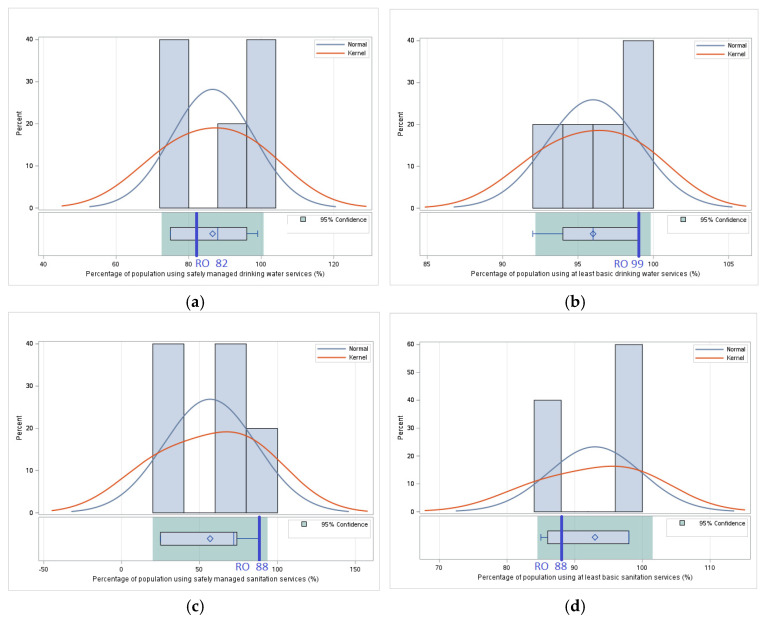
Distribution for values for Pillar IV indicator suite—Utilization—in Romania’s neighboring countries, respectively, the value of the indicator in Romania (RO): (**a**) Percentage of population using safely managed drinking water services (%); (**b**) percentage of population using at least basic drinking water services (%); (**c**) percentage of population using safely managed sanitation services (%); (**d**) percentage of population using at least basic sanitation services (%); (**e**) percentage of children under 5 years of age who are stunted (modeled estimates) (%); (**f**) percentage of children under 5 years of age who are overweight (modeled estimates) (%); (**g**) prevalence of obesity in the adult population (18 years and older) (%); (**h**) prevalence of anemia among women of reproductive age (15–49 years) (%); (**i**) prevalence of low birth weight (%).

**Figure 9 foods-14-01309-f009:**
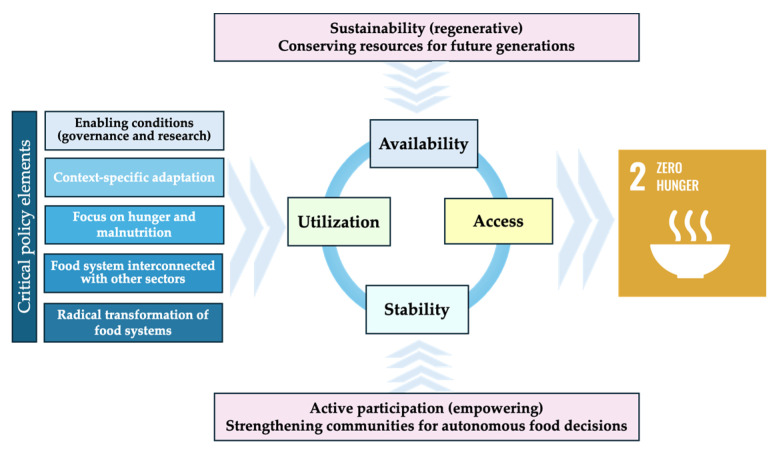
Improving scheme for food security in Romania and neighboring countries. Source: Authors own interpretation by [40].

**Table 1 foods-14-01309-t001:** Food security indicators and reporting years [6].

Pillars and Suites of Indicators	Year/Annual Average from the Range
*Pillar I—Availability*	
Average dietary energy supply adequacy	2021–2023
Share of dietary energy supply derived from cereals, roots, and tubers	2020–2022
Average protein supply	2020–2022
Average supply of protein of animal origin	2020–2022
*Pillar II—Access*	
Rail line density (total route in km per 100 square km of land area)	2021
Gross domestic product per capita (in purchasing power equivalent)	2022
Prevalence of undernourishment, 3-year averages	2021–2023
Prevalence of undernourishment, yearly estimates	2021–2023
Prevalence of severe food insecurity in the total population, 3-year averages	2021–2023
Prevalence of severe food insecurity in the total population, yearly estimates	2021–2023
Prevalence of moderate or severe food insecurity in the total population, 3-year averages	2021–2023
Prevalence of moderate or severe food insecurity in the total population, yearly estimates	2021–2023
*Pillar III—Stability*	
Cereal imports dependency rate	2020–2022
Percent of arable land equipped for irrigation	2020–2022
Value of food imports over total merchandise exports	2020–2022
Political stability and absence of violence/terrorism	2022
Per capita food supply variability	2023
*Pillar IV—Utilization*	
People using at least basic drinking water services	2022
People using safely managed drinking water services	2022
People using at least basic sanitation services	2022
People using safely managed sanitation services	2022
Percentage of children under 5 years of age affected by wasting	2019
Percentage of children under 5 years of age who are stunted	2022
Percentage of children under 5 years of age who are overweight	2022
Prevalence of obesity in the adult population (18 years and over)	2022
Prevalence of anemia in women of reproductive age (15–49 years)	2019
Prevalence of exclusive breastfeeding among infants aged 0–5 months	2019
Prevalence of low birth weight	2020

**Table 2 foods-14-01309-t002:** Values for *Pillar I—Availability* indicators [6].

Indicators of Pillar I—Availability	Bulgaria	Hungary	Moldova	Romania	Serbia	Ukraine
Average dietary energy supply adequacy (%)	121	132	121	146	146	116
Share of dietary energy supply derived from cereals, roots, and tubers (%)	33	28	36	37	49	43
Average protein supply (g/cap/day)	89.0	90.6	88.8	109.3	124.3	91.1
Average supply of protein of animal origin (g/cap/day)	50.4	54.7	42.5	60.1	58.2	45.9

**Table 3 foods-14-01309-t003:** Values for *Pillar II*—*Access* indicators [6].

Indicators of Pillar II—Access	Bulgaria	Hungary	Moldova	Romania	Serbia	Ukraine
Rail line density (total route in km per 100 square km of land area)	3.6	8.6	3.4	4.5	3.9	-
Gross domestic product (GDP) per capita, PPP, (constant 2017 international $)	32,511.8	40,683.9	15,229.5	39,872.5	23,741.2	14,950.5
Prevalence of undernourishment (%)	<2.5	<2.5	<2.5	<2.5	<2.5	5.8
Number of people undernourished (mil.)	-	-	-	-	-	2.3
Prevalence of severe food insecurity in the total population (%)	2.5	3.6	5.3	7.1	3.0	5.3
Prevalence of moderate or severe food insecurity in the total population (%)	14.8	15.0	24.7	19.1	13.0	31.0
Number of severely food-insecure people (mil.)	0.2	0.4	0.2	1.4	0.3	2.1
Number of moderately or severely food-insecure people (mil.)	1.0	1.5	0.8	3.8	1.2	12.4

- *unreported*.

**Table 4 foods-14-01309-t004:** Values of *Pillar III—Stability* indicators [6].

Indicators of Pillar III—Stability	Bulgaria	Hungary	Moldova	Romania	Serbia	Ukraine
Cereal import dependency ratio (%)	−208.0	−90.4	100.0	−81.8	−53.9	−361.9
Percent of arable land equipped for irrigation (%)	1.3	2.1	6.5	13.8	0.8	3.7
Value of food imports in total merchandise exports (%)	11	5	24	12	9	9
Political stability and absence of violence/terrorism (index)	0.3	0.6	−0.7	0.5	−0.2	−2
Per capita food supply variability (kcal/cap/day)	16	49	37	33	39	34

**Table 5 foods-14-01309-t005:** Values of *Pillar IV—Utilization* indicators [6].

Indicators of Pillar IV—Utilization	Bulgaria	Hungary	Moldova	Romania	Serbia	Ukraine
Percentage of population using safely managed drinking water services (%)	96	99	75	82	75	88
Percentage of population using at least basic drinking water services (%)	99	99	92	99	96	94
Percentage of population using safely managed sanitation services (%)	74.0	88.0	-	88.0	25.0	72.0
Percentage of population using at least basic sanitation services (%)	86.0	98.0	85.0	88.0	98.0	98.0
Percentage of children under 5 years affected by wasting (%)	-	-	-	-	2.6	-
Percentage of children under 5 years of age who are stunted (modeled estimates) (%)	5.6		3.9	7.7	4.6	12.3
Percentage of children under 5 years of age who are overweight (modeled estimates) (%)	3.8		2.9	4.5	9.9	13.6
Prevalence of obesity in the adult population (18 years and older) (%)	20.6	31.7	23	34	22.5	23.6
Prevalence of anemia among women of reproductive age (15–49 years) (%)	23.6	19.7	26.1	22.7	22.8	17.7
Prevalence of exclusive breastfeeding among infants 0–5 months of age (%)	-	-	-	-	23.6	-
Prevalence of low birth weight (%)	11.4	8.3	6.5	8.8	6.2	5.7

- *unreported*.

## Data Availability

Data were obtained from the Food and Agricultural Organization of the United Nations (FAO) and are available on the Suite of Food Security Indicators page https://www.fao.org/faostat/en/#data/FS (accessed on 1 September 2024 and 10 January 2025). Metadata regarding the suites of indicators is available at https://www.fao.org/faostat/en/#data/FS/metadata (accessed on 15 September 2024). The software used for statistical analysis *One-sample t-tests* and *Wilcoxon signed-rank tests* are available at https://www.sas.com/ro_ro/software/on-demand-for-academics.html (accessed on 15 September 2024).

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
