# Peer review of "Sustainable Food Security in Romania and Neighboring Countries—Trends, Challenges, and Solutions†"

_foods, 2025, doi:10.3390/foods14081309_

Round 1
Reviewer 1 Report
Comments and Suggestions for Authors
Food security is a fundamental global challenge, which face interconnected challenges. This study assesses Romania’s food security in relation to its neighbors using FAO defined indicators for each of these four pillars. The study has a certain degree of innovation. But I found some problems about it. I do not find some meaning issues about it. Here is my comments.
1.The introduction mentions the impact of Romania and its neighboring countries' communist history on food security, but only briefly refers to "central planning economic policies" and "food rationing" without delving into how these historical factors specifically influence the current food security structure. It is suggested that the specific historical cases or data to illustrate how agricultural policies during the planned economy period led to current agricultural structural issues.
- In the part of Section 2.1 mentions the use of FAO data but does not clearly state the year range and update frequency of the data. For example, some indicators have a large span of years (e.g., 2019-2023), which may lead to inconsistencies in the data.
- In the part of Section 2.1,it ismentions the use of four pillars of food security from the FAO (availability, access, stability, and utilization), but does not explain why these particular indicators were chosen (e.g., the direct relationship between "railway density" and food security).
- In the part of Section 2.2.2 mentions the use of t-tests and Wilcoxon signed-rank tests but does not state the conditions under which these methods are applicable or their limitations.
- In the part ofSection 3.1.1 mentions that Romania has a higher "average dietary energy supply adequacy" than neighboring countries, but does not analyze whether this high supply may lead to resource waste or environmental issues.
- In the part of Section 3.1.2 mentions that Romania has a higher per capita GDP but still has a high rate of food insecurity, but does not delve into how economic inequality affects food security.
- In the part of Section 3.1.2 mentions that Romania has a high railway density, but does not analyze how this infrastructure specifically affects food distribution and accessibility.
- In the part of Section 3.1.3 mentions the potential impact of climate change on food security but does not specifically analyze the measures and challenges that Romania and its neighboring countries face in addressing climate change.
- In the part of Section 4.5 proposes some policy recommendations (such as improving irrigation infrastructure) but does not assess the feasibility and implementation barriers of these suggestions in Romania and its neighboring countries.
10.In the part of Section 3.1.2 mentions that data on "malnutrition incidence" are missing (only Ukraine reported specific values), but does not discuss the impact of these missing data on the study results.
- In the part of Section 4.5 mentions the importance of regional cooperation for food security but does not specifically analyze the current status and challenges of regional cooperation among Romania and its neighboring countries.
12.In the part of Section 3.1.4 mentions that Romania has a high obesity rate but does not delve into the long-term impact of this phenomenon on food security.
- The paper does not mention food waste, which is an important issue in food security.
- The conclusion section only briefly summarizes the main findings of the study without fully discussing their policy implications and future research directions.It is suggested to expand the conclusion section to summarize the main findings in more detail and propose more actionable policy recommendations.
Comments on the Quality of English Language
The English should be polished by native speakers.
Author Response
Dear Reviewer 1,
We would like to express our sincere gratitude for the thorough evaluation of our manuscript and for the insightful and constructive comments provided. We carefully reviewed each observation, and we fully accepted and implemented all the comments and suggestions. The manuscript has been revised accordingly, with modifications applied throughout the entire text. We expanded the explanations, added new references, and improved the clarity and structure of several sections, as recommended. We truly appreciate your contribution to enhancing the quality of our work.
Below are our responses, point-by-point.
Comment 1. The introduction mentions the impact of Romania and its neighboring countries' communist history on food security, but only briefly refers to "central planning economic policies" and "food rationing" without delving into how these historical factors specifically influence the current food security structure. It is suggested that the specific historical cases or data to illustrate how agricultural policies during the planned economy period led to current agricultural structural issues.
Response 1
Thank you for your insightful comment. We have expanded the introduction section to highlight more clearly how agricultural policies from the era of the planned economy have influenced the current agricultural structure and food security. We have included additional references to the impact of land fragmentation, the decline in investment in agricultural infrastructure, and the imbalance between crop and livestock production, issues inherited from the communist era that continue to affect the efficiency and resilience of food systems in Romania and neighboring countries. In the new paragraph added to the introduction, we have mentioned the effects of forced collectivization, regional overspecialization in agricultural production (e.g., state-imposed monocultures), but also the difficulties of the transition after 1989, when land was resold, often in small lots and without technical support, which led to excessive fragmentation and a reduced capacity to adapt to new markets. These aspects partially explain current vulnerabilities in terms of equitable access to food, agricultural competitiveness, and the capacity to respond to crises. We believe these additions bring additional depth and historical contextualization, as you suggested.
Comment 2. In the part of Section 2.1 mentions the use of FAO data but does not clearly state the year range and update frequency of the data. For example, some indicators have a large span of years (e.g., 2019-2023), which may lead to inconsistencies in the data.
Response 2
We agree that the temporal variation in the reporting years of the Food Security Indicators represents an important methodological issue.
To address this issue, we have clarified in the Materials and Methods section that the indicators used in the study are the most recently reported and made available by the FAO. We also emphasize this limitation in the Discussion section (Subsection 4.3 Study limitations and shortcomings), where we explain that the differences in the years of data collection across countries and indicators could affect the comparability of values.
Furthermore, to support our methodological decision, we have contacted the FAOSTAT team directly. They confirmed that the food security data on the FAOSTAT platform were last updated in July 2024 and that, for many indicators, especially those from organizations such as UNICEF and WHO, FAO provides the latest available and reliable estimates, which may refer to previous years or multi-year averages. We have now included this information in the Discussion section to strenghten the validity of the chosen data sources and to explain the rationale behind using indicators with different time frames.
Thank you for your helpful suggestion, which allowed us to further strengthen the transparency and rigor of our methodology.
Comment 3. In the part of Section 2.1,it is mentions the use of four pillars of food security from the FAO (availability, access, stability, and utilization), but does not explain why these particular indicators were chosen (e.g., the direct relationship between "railway density" and food security).
Response 3
It is an insightful comment. Thank you!
We would like to clarify that the indicators used in this study are not self-selected, but are officially defined and published by FAO. These indicators are part of the Food Security Indicators (FSI) database, which is publicly available on the FAOSTAT platform (https://www.fao.org/faostat/en/#data/FS). FAO developed these indicators over decades of international research and expert consultation, with the aim to providing a comprehensive, standardized, and globally recognized framework for assessing food security.
In our study, we included all available and reported indicators corresponding to the four internationally accepted FAO pillars of food security: Availability, Access, Stability, and Utilization. We did not formulate or selectively choose these indicators. For instance, the indicator Railway density is included under the Access pillar by FAO because it is known that transport infrastructure influences physical access to food, especially in rural or disadvantaged areas.
To address to your recommendation, we have now added a clarifying sentence in the Materials and Methods section to explicitly state that all indicators were selected according to the FAO methodology and were used in their entirety as provided in the Food Security Indicators dataset.
Your helpful suggestion allowed us to increase the transparency and scientific foundation of our methodological approach.
Comment 4. In the part of Section 2.2.2 mentions the use of t-tests and Wilcoxon signed-rank tests but does not state the conditions under which these methods are applicable or their limitations.
Response 4
Thank you very much for this pertinent observation. We fully agree that the choice of statistical test must be carefully justified. In our manuscript, we clarified the rationale for applying the One-sample t-test and the Wilcoxon signed-rank test, based on a preliminary normality check using the Shapiro-Wilk test. This normality test is widely used in statistical analysis and is the default method built into the SAS OnDemand for Academics software that we used for this study.
Although normality tests can indeed sometimes misinterpret slight deviations from a normal distribution, we preferred to formally test the data rather than directly apply a parametric test. This decision ensured a more rigorous and transparent approach.
Additionally, we have ensured that this methodological step is explicitly mentioned in the Materials and Methods section of the manuscript as part of our explanation of the statistical analysis process.
We appreciate your comment, which highlights the importance of methodological clarity, and thank you for the opportunity to provide this additional explanation.
Comment 5. In the part Section 3.1.1 mentions that Romania has a higher "average dietary energy supply adequacy" than neighboring countries, but does not analyze whether this high supply may lead to resource waste or environmental issues.
Response 5
We agree that the high value of the Average dietary energy supply adequacy in Romania should not be interpreted exclusively as a positive outcome. A high energy supply may indeed indicate overconsumption, inefficient food systems, or even high levels of food waste — all of which have potential environmental consequences.
In response to your comment, we have revised the Discussion section (Subsection 4.1, related to Pillar I: Availability) to include a paragraph reflecting these concerns. Specifically, we now state that the high dietary energy intake should be analyzed together with nutritional quality, food waste potential, and environmental sustainability, and that future studies should investigate these connections in more depth.
We appreciate your suggestion, which allows us to broaden the interpretative scope of our findings.
Comment 6. In the part of Section 3.1.2 mentions that Romania has a higher per capita GDP but still has a high rate of food insecurity, but does not delve into how economic inequality affects food security.
Response 6
Thank you for your observation. Our analysis already included this perspective – namely that a high GDP per capita at the national level can hide major structural vulnerabilities and social inequalities that influence actual access to food. This was mentioned in the subsection “Pillar II – Accessibility” in section 4.1. Comparative Analysis – Main Findings.
However, following your suggestion, we found that the previous wording could be interpreted as too general, so we have reworded the passage to more clearly highlight the link between economic inequality and food insecurity. Indeed, the new version we have formulated was necessary and includes explicit references to income distribution, unequal access to food in rural areas, and the effects of inflation on vulnerable households. We hope that this rewording fully responds to your suggestion and contributes to an important conceptual clarification.
Comment 7. In the part of Section 3.1.2 mentions that Romania has a high railway density, but does not analyze how this infrastructure specifically affects food distribution and accessibility.
Response 7
Thank you for your observation. I mentioned the railway density indicator in the comparative context of the “Access” pillar, because it is included in the official set of indicators defined by the FAO for assessing food security. As you pointed out, a simple numerical mention is not enough to reflect the real role of this type of infrastructure in access to food. Following your suggestion, we have completed the text to include a brief qualitative analysis on how the railway network in Romania, although extensive in numbers, does not necessarily translate into efficient or equitable distribution of food. We have highlighted the structural problems and territorial inequalities that limit the capacity of railway infrastructure to contribute significantly to reducing food insecurity, especially in rural areas.
Comment 8. In the part of Section 3.1.3 mentions the potential impact of climate change on food security but does not specifically analyze the measures and challenges that Romania and its neighboring countries face in addressing climate change.
Response 8
We fully agree that climate change, as a key factor influencing food security, deserves to be addressed more explicitly in the paper. Although Section 3.1.3 mentioned climate risks in a general way, we have responded to your suggestion by adding a dedicated paragraph in the Introduction section.
This additional paragraph details the specific climate challenges facing Romania and its neighboring countries (such as prolonged droughts, extreme weather events, and low investment in modern irrigation systems) and highlights regional differences in adaptive capacity.
We chose to include this addition in the introductory section rather than in 3.1.3 because we felt it provided a more appropriate conceptual framework to understand the systemic dimension of climate change and its impact on all four pillars of food security. In this way, we highlight the need for coordinated and resilient climate policies in the region right from the outset of the paper.
Thank you for this observation, which helped us improve the clarity and contextual relevance of the study.
Comment 9. In the part of Section 4.5 proposes some policy recommendations (such as improving irrigation infrastructure) but does not assess the feasibility and implementation barriers of these suggestions in Romania and its neighboring countries.
Response 9
We appreciate your observation regarding the feasibility and challenges of implementing the recommendations formulated in Section 4.5. Indeed, we proposed a series of measures, such as improving irrigation infrastructure, developing integrated strategies to combat food insecurity or reducing regional disparities, without detailing the concrete obstacles that may arise in the implementation process.
Following this observation, we have supplemented Section 4.5 with a text stating that the implementation of these measures is conditional on factors such as: institutional capacity, the level of available investments, the degree of coordination between public authorities and the private sector, as well as the support of the population. We have also emphasized that there are significant differences between the countries analyzed in terms of resources and strategic priorities, and this must be taken into account in the formulation of public policies.
Comment 10. In the part of Section 3.1.2 mentions that data on "malnutrition incidence" are missing (only Ukraine reported specific values), but does not discuss the impact of these missing data on the study results.
Response 10
Thank you for your observation. Indeed, we mentioned in section 3.1.2 that only Ukraine reported values for the indicator Number of people undernourished (mil.), which limited the possibility of conducting a relevant comparative analysis between the six countries. We appreciate your remark and have now completed the text in section 3.1.2 with a clear mention that the impact of the lack of reported values for this indicator will be discussed in section 4.3 – Study limitations and shortcomings. We believe that this addition brings added coherence and transparency to the methodological presentation of the study.
Comment 11. In the part of Section 4.5 mentions the importance of regional cooperation for food security but does not specifically analyze the current status and challenges of regional cooperation among Romania and its neighboring countries.
Response 11
Thank you for your observation. Indeed, in section 4.5 we highlighted the importance of regional cooperation for improving food security, but without detailing the current state and specific challenges of cooperation between Romania and neighboring countries. Your observation is very valuable and, following it, we added a sentence to the text that recognizes this limitation and emphasizes the need for future research that explores in depth the dimension of regional cooperation in the field of food security. We believe that this addition brings clarity and opens an important direction for the development of future public policies.
Comment 12. In the part of Section 3.1.4 mentions that Romania has a high obesity rate but does not delve into the long-term impact of this phenomenon on food security.
Response 12
Thank you for your observation. We fully agree that the high obesity rate in Romania is not only a public health problem, but also a factor affecting food security in the long term. Obesity reflects imbalances in access to a balanced and healthy diet and frequently indicates an increased consumption of ultra-processed foods, rich in calories but poor in essential nutrients. This reality directly affects the Utilization pillar of food security, negatively influencing population health, healthcare system costs and economic productivity.
We have completed section 4.1. – Comparative analysis – Main findings (Pillar IV – Utilization), adding a reflection on the long-term implications of obesity on food security. We believe that it is essential that food policies aim not only at the availability and access to food, but also at the promotion of healthy and sustainable food patterns, which reduce the incidence of chronic diseases and support the health of future generations.
Comment 13. The paper does not mention food waste, which is an important issue in food security.
Response 13
Thank you for your very pertinent observation regarding the absence of an explicit discussion of food waste, a key topic in the analysis of food security. We have integrated this recommendation into the manuscript, including a new paragraph in section 4.1 – Pillar IV: Utilization, which highlights both food loss (at the production, processing and distribution stages) and food waste (especially at the level of consumers, the HoReCa sector and public institutions).
We highlighted the negative impact of these phenomena on food availability, resource sustainability and the environment, as well as the need to include measures to prevent and reduce food losses and waste in public food security policies.
This addition strengthens the holistic nature of the analysis and reflects a coherent approach to all dimensions relevant to a sustainable food system.
Comment 14. The conclusion section only briefly summarizes the main findings of the study without fully discussing their policy implications and future research directions.It is suggested to expand the conclusion section to summarize the main findings in more detail and propose more actionable policy recommendations.
Response 14
Thank you for this extremely important observation. We have completely revised the Conclusions section, which we have expanded to reflect in detail the main findings of the study for each of the four pillars of food security. We have also included specific policy recommendations for each pillar, based on the results obtained, as well as direct references to future research directions. We have also integrated into this section the analysis of the link between the study results and several Sustainable Development Goals (SDGs), providing an integrative perspective on the long-term implications. We believe that the new version of the Conclusions fully responds to your suggestion.
Reviewer 2 Report
Comments and Suggestions for Authors
In the introduction, it is necessary to add the following points:
priority foods, production, consumption, economic impact.
L65-67 What has been the impact of climate change on food production in Romania and neighboring countries? It is necessary to add research on the effects caused over the last 20 years.
L88 Which crops are the most important? How do they impact the food gross domestic product? Sources of employment, number of production systems?
L114-124 This entire section needs to be rewritten and the information properly organized so that it ends with the objective, written clearly and consistent with the title.
L127-137 How many articles were analyzed? From what years? Aside from keywords, what other criteria were used for article selection?
L150-152 How many documents were considered for each pillar? L155 Most recent refers to 2024?
L164 What does the regional average refer to?
L208-209 What was the value of 2.5 based on?
L293-302 Indicate what are the benefits of having greater availability of animal protein? Are people's daily consumption requirements met? How do production systems operate in the generation of animal protein?
In the discussion, it is necessary to increase the contrast with other research that allows for greater evidence of food security in the context of the study and other contexts such as Latin America, Asia, and North America. Mention the possible public policies that need to be implemented to improve food security?
The figures should have a higher resolution.
What are the limitations of the research?
Author Response
Dear Reviewer 2,
We would like to express our sincere gratitude for the time and effort dedicated to reviewing our manuscript and for the constructive comments provided. We carefully analyzed each suggestion and revised the manuscript accordingly. In all cases, we either modified or supplemented the content based on the observations received, or clearly indicated where in the text the requested information was already included.
We truly appreciate your contributions, which have helped us improve the clarity, depth, and relevance of our study, and we are confident that the revised version of the manuscript reflects these enhancements.
Below are our responses, point-by-point.
Comment 1. In the introduction, it is necessary to add the following points: priority foods, production, consumption, economic impact.
Response 1 Thank you very much for your suggestion. We fully agree that a more detailed contextualization of priority foods, production, consumption patterns, and their economic impact adds valuable depth to the Introduction. Therefore, we have now included a new paragraph in the final part of the Introduction, which briefly discusses the main food products in the region, the importance of agricultural production in the economy, current consumption trends, and their implications for both public health and food system sustainability.
Comment 2. L65-67 What has been the impact of climate change on food production in Romania and neighboring countries? It is necessary to add research on the effects caused over the last 20 years.
Response 2 Thank you for this important observation. We agree that understanding the impact of climate change on food production is essential in the context of food security. In response to your comment, we have enriched the Introduction with an additional paragraph summarizing existing research on the effects of climate change over the past two decades in Romania and neighboring countries. This paragraph discusses the increased frequency of droughts, temperature fluctuations, and extreme weather events that have affected agricultural productivity and stability in the region. These insights help to better position our study within the broader climate-related vulnerabilities that influence food systems.
Comment 3. L88 Which crops are the most important? How do they impact the food gross domestic product? Sources of employment, number of production systems?
Response 3 Thank you for your valuable comment. In response, we have revised the text to include specific details on Romania’s main agricultural crops (wheat, maize, sunflower, and barley) and their contribution to the national economy. We clarified that agriculture accounts for approximately 4% of Romania’s GDP (according to Eurostat and the National Institute of Statistics, 2022), which is above the EU average. Furthermore, we have highlighted that around 20% of the rural active population is employed in the agricultural sector, underlining its role in rural cohesion and employment. We have also described the dual structure of production systems in Romania, consisting of both small-scale, poorly mechanized family farms and large commercial farms integrated into international supply chains. These additions provide a more comprehensive understanding of the Romanian agri-food system, as suggested.
Comment 4. L114-124 This entire section needs to be rewritten and the information properly organized so that it ends with the objective, written clearly and consistent with the title.
Response 4 Thank you for your observation. We have completely reworded the paragraph in lines 114–124 to make it clearer, more logically structured, and in full accordance with the title of the paper. The new text explicitly defines the main objective of the research — the comparative assessment of food security in Romania and neighboring countries using the four FAO pillars — as well as the secondary objectives, relating to the formulation of recommendations for sustainable public policies and strengthening regional cooperation. We have also eliminated ambiguous formulations and focused the message on the regional and scientific relevance of the study, in the current context of climatic, economic and geopolitical challenges.
Comment 5. L127-137 How many articles were analyzed? From what years? Aside from keywords, what other criteria were used for article selection?
Response 5 Thank you for your pertinent observation. We have completed the corresponding section (lines 127–137) with additional information on the literature selection methodology: we have specified that 57 papers published between 2005 and 2023 were analyzed, and in addition to the use of keywords, additional criteria were applied, such as relevance to the four pillars of food security, regional applicability, and scientific or institutional recognition of the source. These additions clarify the scientific substantiation of the analysis and respond to the requirement of methodological transparency.
Comment 6. L150-152 How many documents were considered for each pillar? L155 Most recent refers to 2024?
Response 6 Thank you for your comments. In response, we have clarified in the manuscript that the analysis included a total of 21 statistical indicators, distributed across the four pillars of food security defined by FAO as follows: 4 indicators for Pillar I – Availability, 4 indicators for Pillar II – Access, 5 indicators for Pillar III – Stability, and 8 indicators for Pillar IV – Utilization. We have also specified that we used the most recent values available in the FAOSTAT database, last updated in July 2024. Given the variation in data availability, the reporting year may differ for each indicator and country. This issue has also acknowledged as a study limitation in Section 4.3.
Comment 7. L164 What does the regional average refer to?
Response 7. Thank you for the comment. As detailed in Section 2.2.1 – Comparative analysis, the regional average refers to the arithmetic mean of the indicator values for the five neighboring countries analyzed: Bulgaria, Hungary, Serbia, Ukraine, and the Republic of Moldova. The calculation method is explicitly described using the arithmetic average formula, and this definition is consistently applied throughout the analysis.
Comment 8. L208-209 What was the value of 2.5 based on?
Response 8. Thank you for this observation. The notation "<2.5%" refers to how FAO and related international organizations (such as WHO or UNICEF) report the indicator Prevalence of Undernourishment when the value is very low and cannot be estimated accurately. This reporting convention is not defined by the authors, but by the data providers themselves and is used when statistical uncertainty prevents precise quantification.
In our study, such indicators were excluded from comparative or statistical analysis because the lack of exact numerical values made it impossible to include them consistently. This is explained in Section 3.1.2 and further discussed in Section 4.3 – Study limitations and shortcomings.
Comment 9. L293-302 Indicate what are the benefits of having greater availability of animal protein?
Response 9. Thank you for your observation. We agree with the relevance of this point and have revised the paragraph to better emphasize the nutritional benefits of higher availability of animal-based protein. In the updated version, we highlight its contribution to the intake of essential amino acids, iron, and vitamin B12—particularly important for vulnerable groups such as children and pregnant women.
Comment 10. Are people's daily consumption requirements met? How do production systems operate in the generation of animal protein?
Response 10. Thank you for your observation. In Section 4.1 of the manuscript, we have already discussed the implications of the increased availability of animal-based protein in Romania, including its nutritional value and the sustainability challenges associated with livestock farming. However, to further strengthen the explanation, we have now completed this paragraph by adding a clarification on how animal protein is generated in Romania. Specifically, we mention the dual character of Romania’s agricultural system, composed of small-scale family farms oriented toward subsistence and large-scale commercial operations integrated into national and export markets. This addition offers a more comprehensive understanding of the structural basis for animal protein production in Romania.
Comment 11. In the discussion, it is necessary to increase the contrast with other research that allows for greater evidence of food security in the context of the study and other contexts such as Latin America, Asia, and North America. Mention the possible public policies that need to be implemented to improve food security.
Response 11. Thank you very much for this thoughtful and constructive suggestion. In response, we have revised the final part of Section 4.5 to include a broader comparative perspective, introducing references to other international experiences in addressing food security challenges in regions such as Latin America, Asia, and North America. Specifically, we have added examples such as Brazil’s Fome Zero program, India’s Public Distribution System (PDS), and the United States' SNAP and community-supported agriculture initiatives, which illustrate how different countries have implemented diverse public policy solutions to combat hunger and food insecurity.
We believe this expanded discussion enhances the international relevance of our findings and aligns with the global scope of food security research. Furthermore, we have integrated a reference to SDG 17 – Partnerships for the Goals, emphasizing the importance of international cooperation and knowledge exchange in achieving sustainable and resilient food systems. The revised paragraph is included at the end of Section 4.5 (Discussion)
Comment 12. The figures should have a higher resolution.
Response 12 Although the figures appear in lower resolution in the PDF version, they were submitted separately in high-resolution formats, as required by the journal.
Comment 13. What are the limitations of the research?
Response 13 We have dedicated an entire subsection, Section 4.3 – Study Limitations and Shortcomings, to discuss the limitations of our research in detail.
Reviewer 3 Report
Comments and Suggestions for Authors
Overall, the manuscript is very well written and highlights the major aspects and impact of food chain supply. I recommend publishing this article with minor revisions. I have the following comments needed to be addressed
Comment-1-Line 72-74- Please highlight this information in in-depth with relevant examples.
Comment-2-Line 79-85- Is there any authentic statistical data to highlight the situation of food supply in Romania before and after the Ukraine war?
Comment-3-Line 240-243-Please do the correction of font.
Comment-4- Line 240-250-Please provide some valid scientific references for this statement.
Comment-5-Also, regarding the statement of lines 281-283, please give an example of previously reported data witha citation of some valid reference to justify this statement.
Comment-6-In the result section, if authors are trying to give some conclusion from their results, it is mandatory to cite previously published research to support their argument. Please do this throughout the manuscript.
Comment-7- Please make the legend of all figures and tables more descriptive. Also, please correct the error in the figure 3. Also, remove gridlines from all figures. Tables 3 and 5 should be reconstructed.
Comment-8- It is suggested to make the conclusion section short and concise.
Comments on the Quality of English Language
Could be improved
Author Response
Dear Reviewer 3,
We sincerely thank you for your thorough review, insightful comments, and valuable suggestions. Your feedback has been extremely helpful in improving the clarity, coherence, and overall quality of our manuscript. We have gratefully accepted all your recommendations and implemented the necessary changes throughout the text.
The only exception concerns your comment regarding the font style used for the names of the indicators (Comment 3). In this case, we provided a detailed justification for our formatting choice, which was intended to ensure clarity and consistency across the manuscript. However, we remain fully open to adapting it according to the editorial guidelines of the Foods journal, should this be required.
Once again, we are truly grateful for your time and constructive input, which have contributed significantly to strengthening our study.
Below are our responses, point-by-point.
Comment-1-Line 72-74- Please highlight this information in in-depth with relevant examples.
Response 1
Thank you for your valuable observation. We agree that the sentence referring to the transition to a market economy required further elaboration. We have now expanded this part of the introduction by providing concrete examples of how the structural changes since 1990 have influenced the current food security system in Romania. Specifically, we mention the fragmentation of agricultural land following property restitution and the dismantling of the centralized food distribution network, which led to disparities in market access between urban and rural areas. These examples help contextualize persistent regional inequalities and structural vulnerabilities.
Comment-2-Line 79-85- Is there any authentic statistical data to highlight the situation of food supply in Romania before and after the Ukraine war?
Response 2
Thank you for your observation regarding the need for statistical data that would highlight the food supply situation in Romania before and after the war in Ukraine. This is indeed an important point. However, official databases such as FAOSTAT do not organize their data based on geopolitical events, but rather report indicators on a standard annual basis. Consequently, there are no disaggregated or comparative food supply indicators published specifically for "pre-war" versus "post-war" periods.
In our study, we used the most recent available data for each indicator, covering the reporting years 2019 to 2023. These reflect, at least in part, the regional consequences of the war, particularly in terms of supply chain instability, rising food price, and import dependency. To ensure accuracy, we contacted the FAOSTAT support team and received official confirmation that the platform was last updated in July 2024 and the latest values reflect the latest available data from both FAO and associated institutions (such as UNICEF and WHO). The email confirmation clearly stated:
“I would like to assure you that our data under FAOSTAT https://faostat.fao.org/internal/en/#data/FS have been updated in July 2024. This implies that there is no data on 2024 but the latest possible available data for all indicators is for 2023.
Note that our platform does not provide updated data up to 2023 for all indicators, basically because of data scarcity and due to the fact that many of the indicators you listed below are actually driven form other organizations’ databases like UNICEF and WHO.”
We have now added a clarifying sentence in the Introduction to explain this limitation and to note that the post-war effects are partially captured by the latest reported values. We hope this additional explanation addresses your comment and enhances the transparency of our methodology.
Comment-3-Line 240-243-Please do the correction of font.
Response 3
Thank you for your observation. We have carefully checked the indicated portion and confirm that the text written in italics refers to the official names of the indicators used in the study. We chose to render them in italics precisely to clearly distinguish them from the rest of the text, thus ensuring readability and coherence. This formatting convention has been applied consistently throughout the entire work, to maintain stylistic unity and to facilitate the identification of the indicators within the four pillars of food security.
We also note that this formatting choice can be adjusted according to final editorial requirements, if necessary and requested by the editors, and we are open to adapting it in accordance with the editorial style of the Foods journal.
Comment-4- Line 240-250-Please provide some valid scientific references for this statement.
Response 4
Thank you very much for this observation. We have carefully reviewed the paragraph in question and agree that a clarification was needed. In the current version of the revised manuscript, we have emphasized that the indicators Number of people with severe food insecurity and Number of people with moderate or severe food insecurity were not included in the comparative analysis because they express the same phenomenon as Prevalence of severe food insecurity in the total population and Prevalence of moderate or severe food insecurity in the total population, but in absolute numbers.
Comparing absolute numbers from countries with very different population sizes can lead to distorted interpretations. This is why prevalence (%) is internationally recommended for cross-country analyses. As outlined in FAO, The State of Food Security and Nutrition in the World 2023, prevalence is the standard metric used to allow for accurate and comparable assessments of levels of food insecurity across countries and regions.
Thus, using prevalence ensured methodological consistency and removed the influence of population size from our comparative analysis. We have kept this approach consistently throughout the study and have now clarified this rationale more explicitly in the manuscript.
Comment-5-Also, regarding the statement of lines 281-283, please give an example of previously reported data with a citation of some valid reference to justify this statement.
Response 5
Thank you for the observation. We have now added a relevant citation to support the statement:
Balan, I.M.; Gherman, E.D.; Brad, I.; Gherman, R.; Horablaga, A.; Trasca, T.I. Metabolic Food Waste as Food Insecurity Factor—Causes and Preventions. Foods 2022, 11(15), 2179.
Comment-6-In the result section, if authors are trying to give some conclusion from their results, it is mandatory to cite previously published research to support their argument. Please do this throughout the manuscript.
Response 6
Thank you for your observation. In response to the comments received from all three reviewers, the manuscript has been thoroughly revised in its entirety. As part of this comprehensive update, we have significantly strengthened the academic foundation of the study by adding numerous bibliographic references—both previous works authored by the research team and relevant, recent studies from other scholars in the field.
These references have been carefully selected and integrated throughout the Results and Discussion sections to support the interpretations derived from the data and to align the arguments with the current scientific literature. We have ensured that we provide an academic basis for the main findings, methodological decisions, and contextual analyses, as also illustrated in our response to Comment 5.
Comment-7- Please make the legend of all figures and tables more descriptive. Also, please correct the error in the figure 3. Also, remove gridlines from all figures. Tables 3 and 5 should be reconstructed.
Response 7
Thank you for your observation. Following your suggestion, we have corrected Figure 3 and removed gridlines from all figures to improve visual clarity. Additionally, as part of our effort to enhance consistency and structure, we have reconstructed Tables 3 and 5 and also revised Tables 2 and 4 to align them with the new format. This ensures that all tables in the manuscript now follow a similar and coherent design, facilitating easier comparison and interpretation of data across the four pillars of food security.
Comment-8- It is suggested to make the conclusion section short and concise.
Response 8
Thank you very much for your suggestion regarding the structure and length of the conclusion section. We would like to mention that one of the reviewers provided a divergent recommendation, explicitly stating.
In response to both comments, we have carefully revised the Conclusions section, aiming to address both perspectives as fully and thoughtfully as possible. While we have maintained a more concise and structured narrative—avoiding repetition and overly technical language—we have also extended the section to include detailed, targeted, and actionable policy recommendations based on the main findings of the study. These recommendations are clearly linked to the four pillars of food security (as defined by FAO) and to the specific challenges identified for Romania and neighboring countries.
Furthermore, the Conclusions now integrates a deeper reflection on the implications of the results in relation to the Sustainable Development Goals (SDGs), in particular SDG 2 (Zero Hunger), SDG 1 (No Poverty), SDG 3 (Good Health and Well-being), SDG 6 (Clean Water and Sanitation), SDG 12 (Responsible Consumption and Production), and SDG 13 (Climate Action).
Our intention was to ensure that the Conclusions section serves not only as a summary, but also as a bridge between research and policy action—highlighting the relevance of our findings and providing a clear direction for future interventions and studies.
We hope this revised version meets your expectations, while reflecting the comprehensive feedback received from both reviewers.
Round 2
Reviewer 1 Report
Comments and Suggestions for Authors
The manuscript has been modified according to the comments. I agree it can be accepted.
Comments on the Quality of English Language
The English should be polished by native speakers. There are some problems about the English., such as lines 456-457.